# Improving the Accuracy of Analytical Relationships for Mechanical Properties of Permeable Metamaterials

**Reza Hedayati [1],\*, Naeim Ghavidelnia [2], Mojtaba Sadighi [2] and Mahdi Bodaghi [3]**

[1] Department of Aerospace Structures and Materials, Faculty of Aerospace Engineering, Delft University of Technology (TU Delft), Kluyverweg 1, 2629 HS Delft, The Netherlands

[2] Department of Mechanical Engineering, Amirkabir University of Technology (Tehran Polytechnic), Hafez Ave, Tehran 15875-4413, Iran; n.ghavidelnia@aut.ac.ir (N.G.); mojtaba@aut.ac.ir (M.S.)

[3] Department of Engineering, School of Science and Technology, Nottingham Trent University, Nottingham NG11 8NS, UK; mahdi.bodaghi@ntu.ac.uk

\* Correspondence: r.hedayati@tudelft.nl

**Abstract:** Permeable porous implants must satisfy several physical and biological requirements in order to be promising materials for orthopaedic application: they should have the proper levels of stiffness, permeability, and fatigue resistance approximately matching the corresponding levels in bone tissues. This can be achieved using designer materials, which exhibit exotic properties, commonly known as metamaterials. In recent years, several experimental, numerical, and analytical studies have been carried out on the influence of unit cell micro-architecture on the mechanical and physical properties of metamaterials. Even though experimental and numerical approaches can study and predict the behaviour of different micro-structures effectively, they lack the ease and quickness provided by analytical relationships in predicting the answer. Although it is well known that Timoshenko beam theory is much more accurate in predicting the deformation of a beam (and as a result lattice structures), many of the already-existing relationships in the literature have been derived based on Euler–Bernoulli beam theory. The question that arises here is whether or not there exists a convenient way to convert the already-existing analytical relationships based on Euler–Bernoulli theory to relationships based on Timoshenko beam theory without the need to rewrite all the derivations from the start point. In this paper, this question is addressed and answered, and a handy and easy-to-use approach is presented. This technique is applied to six unit cell types (body-centred cubic (BCC), hexagonal packing, rhombicuboctahedron, diamond, truncated cube, and truncated octahedron) for which Euler–Bernoulli analytical relationships already exist in the literature while Timoshenko theory-based relationships could not be found. The results of this study demonstrated that converting analytical relationships based on Euler–Bernoulli to equivalent Timoshenko ones can decrease the difference between the analytical and numerical values for one order of magnitude, which is a significant improvement in accuracy of the analytical formulas. The methodology presented in this study is not only beneficial for improving the already-existing analytical relationships, but it also facilitates derivation of accurate analytical relationships for other, yet unexplored, unit cell types.

**Keywords:** mechanical properties; Euler–Bernoulli beam theory; Timoshenko beam theory; analytical relationship; finite element method

## 1. Introduction

Recently, partially or fully porous load-bearing implants have been proposed to replace the traditional solid implants for repairing large bony defects. While metallic foams manufactured by conventional techniques such as powder metallurgy [1], investment casting [2], and space-holder [3,4] have found their way in this field [5], they all lack a good controllability over the microstructural geometry of the implants, and hence their

static mechanical properties [6], fatigue resistance [7], and biological response [8]. A recent explosion in the application of additive manufacturing (AM) in biomedical engineering has opened the possibility of manufacturing porous meta-implants with arbitrary micro-architecture. AM makes it possible to manufacture open-cell (i.e., permeable) porous materials with precisely designed microstructure both in micro- and macro-scales [9].

Porous implants should satisfy several physical and biological requirements in order to be in an optimal state for biomedical applications: they should have the right levels of stiffness, permeability, and fatigue resistance, in proximity to how much they are in bone tissues. This is especially crucial to avoid the undesired consequences of using highly stiff solid metallic implants that can cause problems such as stress shielding [10]. This can be achieved using designer materials, which can exhibit exotic properties, commonly known as metamaterials. Permeable metamaterials have shown several advantages in multi-functional applications such as biomedical engineering, acoustics, photonics, and thermal management [11].

Metamaterials used to construct implants are made of repeating building blocks known as unit cells. The mechanical, physical, and biological properties of implants are determined by four main characteristics of the unit cell that they are made of: the shape of the cells, their size, their permeability, and their relative density (which is defined as the fraction of space occupied by the solid material).

In recent years, several experimental [12–19], numerical [20–25], and analytical [26–30] studies have been dedicated to studying the influence of unit cell shape on the above-mentioned properties. Even though experimental and numerical approaches can study and predict the behaviour of different micro-structures effectively, they lack the ease and quickness provided by analytical relationships in predicting the answer. Analytical relationships for a regularly repeated lattice structure have several benefits: they can be used for validation of numerical or experimental results, they can be implemented in Machine Learning or Artificial Intelligence algorithms to construct optimally designed patient-specific porous implants, and they can give a clear and quick indication of what geometrical/material properties have the most contribution in each of the mechanical properties.

Previously, analytical relationships for elastic properties (elastic modulus, Poisson's ratio, and yield stress) of different unit cells such as body-centred cubic (BCC) [31], cube [32,33], diamond [26,34], hexagonal packing [35], iso-cube [5], octahedral [27], rhombic dodecahedron [28,29], rhombicuboctahedron [30], truncated cube [32], truncated cuboctahedron [36], and truncated octahedron [37] have been derived. In the literature, relationships for octahedral [27], rhombic dodecahedron [29], and truncated cuboctahedron [36] have been derived based on both Euler–Bernoulli and Timoshenko beam theories. However, for the rest of the above-mentioned geometries, the relationships are derived based on Euler–Bernoulli beam theory only. The question that arises here is whether or not there can be a convenient way to convert the already-existing relationships based on Euler–Bernoulli to relationships based on Timoshenko beam theory without the need to rewrite all the derivations from the starting point.

In this study, we try to answer that question by presenting a technique to convert the already-existing analytical relationships based on Euler–Bernoulli beam theory to equivalent Timoshenko ones. This technique can be used to convert Euler–Bernoulli elastic relationships including elastic modulus, Poisson's ratio, and yield stress ($E, \sigma_y, \nu$) of any porous material. The efficiency of this technique is evaluated for six unit cell types: BCC, hexagonal packing, rhombicuboctahedron, diamond, truncated cube, and truncated octahedron. Numerical simulations are also carried out using finite element (FE) modelling to evaluate whether or not the new technique enhances the accuracy of the analytical results. Moreover, experimental data points from previous studies are used for validation of the proposed technique.

## 2. Materials and Methods

The general deformation of any arbitrary strut of an open-cell lattice structure can be decomposed into four basic deformations (Figure 1). Each strut in the unit cell could be considered as a clamped beam with four main deformations namely lateral displacement ($v$), flexural rotation ($\theta$), twist ($\varphi$), and elongation/contraction ($u$). By considering these deformations as the basic deformation modes of the struts of a lattice structure, the total deformation and behaviour of a unit cell in a lattice structure could be calculated and obtained. In Sections 2.1 and 2.2, the basic formulation of deformation for a single strut under various conditions are obtained based on two well-known beam theories namely Euler–Bernoulli and Timoshenko (For detailed description please see Appendix A).

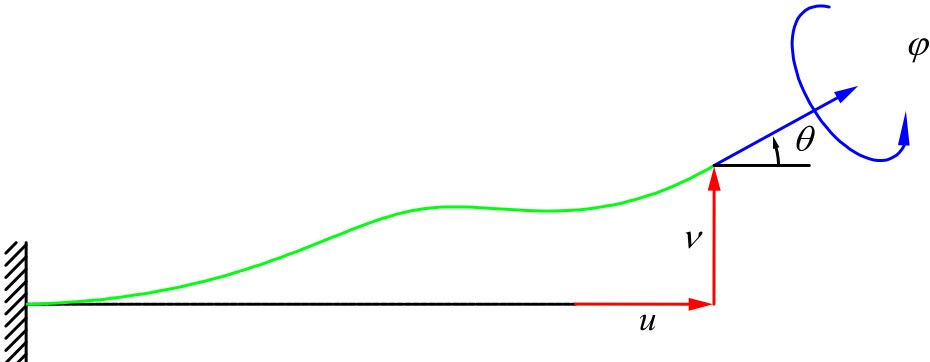

**Figure 1.** General deformation of a cantilever beam with axial and lateral displacements as well as flexural and torsional rotations at the free end.

### 2.1. Euler–Bernoulli Beam Theory

For a beam under no distributed load, the Euler–Bernoulli beam equation can be written as

$$\frac{d^4 w}{dx^4} = 0 \tag{1}$$

where $w$ is the deflection. The solution to this differential equation can be expressed as:

$$w = c_0 + c_1 x + c_2 x^2 + c_3 x^3 \tag{2}$$

where constants, $c_0 - c_3$, are determined by applying boundary conditions. For a cantilever Euler–Bernoulli beam with a point load, $P$, at its end, we have

$$\delta = \frac{Fl^3}{3E_s I} \qquad and \qquad \theta = \frac{Fl^2}{2E_s I} \tag{3}$$

On the other hand, for a cantilever beam with a concentrated moment, $M$, at its end, the displacement and rotation are as follows:

$$\delta = \frac{Ml^2}{2E_s I} \qquad and \qquad \theta = \frac{Fl}{E_s I} \tag{4}$$

In beams where the angle of the free end does not change during the deformation (e.g., the beams of the lattice structure considered in this study), the rotations produced by the lateral load, $F$, and moment, $M$, must be equal and opposite, from which the value of $M$ can be identified:

$$\frac{Fl^2}{2E_s I} = \frac{Ml}{E_s I} \qquad \rightarrow \qquad M = \frac{Fl}{2} \tag{5}$$

While force, $F$, tends to increase the deflection, moment, $M$, tends to reduce it. The total deflection created by force, $F$, and moment, $M$, is then

$$\delta = \frac{Fl^3}{3E_sI} - \left(\frac{Fl}{2}\right)\frac{l^2}{2EI} = \frac{Fl^3}{12EI} \tag{6}$$

Rewriting Equation (6) as a function of $F$ gives (see Figure 2a)

$$F = \frac{12E_sI}{l^3}\delta \tag{7}$$

According to Equation (5), we will have $M = \frac{6E_sI}{l^2}\delta$. Similarly, the axial force required to displace the end of a rod for $u$ is $AE_su/l$ (see Figure 2c). The equations for a cantilever beam with rotation but with no displacement in the end can be obtained in a similar manner and the results are shown in Figure 2b.

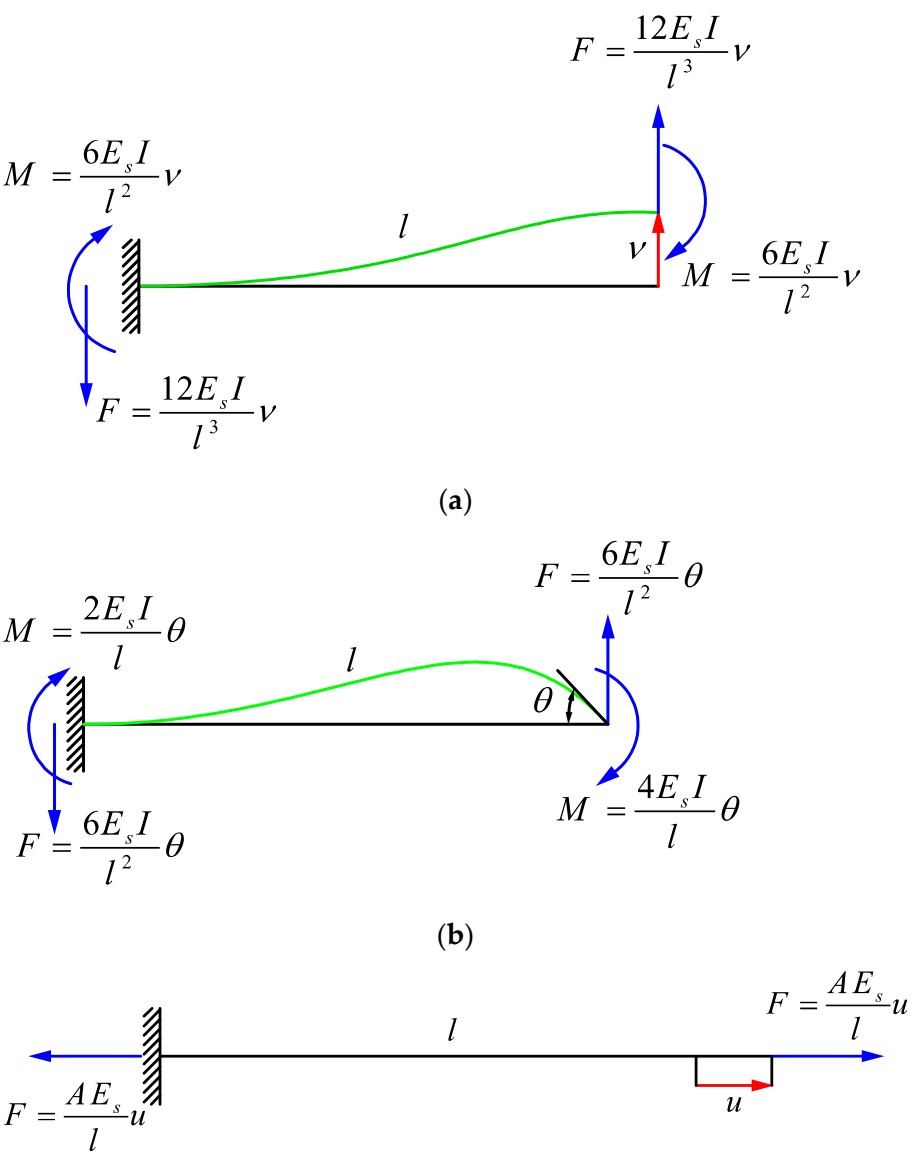

(**a**)

(**b**)

(**c**)

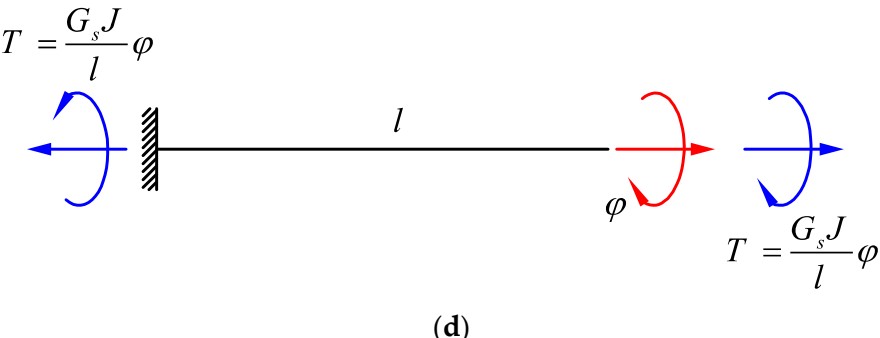

(**d**)

**Figure 2.** Forces and moments required to cause (**a**) lateral displacement, $\delta$, with no rotation at the free end of the beam, (**b**) rotation, $\theta$, with no lateral displacement at the free end of the beam, (**c**) pure axial extension, and (**d**) pure twist in the free end of an Euler–Bernoulli beam [32,36].

*2.2. Timoshenko Beam Theory*

　　Now, we try to find out how the moments and forces shown in Figure 2a,b would change due to change in the beam theory. The Timoshenko beam theory takes into account shear deformation and rotational inertia effects, making it suitable for describing the behaviour of short beams. For a homogenous beam of constant cross-section, the Timoshenko beam governing equations are as follows:

$$\frac{d^2}{dx^2}\left(E_sI\frac{d\varphi}{dx}\right) = q(x,t)$$

$$\frac{dw}{dx} = \varphi - \frac{1}{\kappa AG_s}\frac{d}{dx}\left(E_sI\frac{d\varphi}{dx}\right)$$

(8)

where $\varphi$ is the angle of rotation of the normal to the mid-surface of the beam and $\kappa$ is the shear coefficient factor. The coefficient, $\kappa$, is a dimensionless quantity, dependent on the shape of the cross-section, which is introduced to account for the fact that the shear stress and strain are distributed not-uniformly over the cross-section [38]. In a linear elastic Timoshenko beam, the bending moment, $M_{xx}$, and the shear force, $Q_x$, are related to the angle of rotation, $\varphi$, and the deflection, $w$, by

$$M_{xx} = -E_sI\frac{\partial\varphi}{\partial x}$$

$$Q_x = \kappa AG_s(-\varphi + \frac{\partial w}{\partial x})$$

(9)

　　(a) A cantilever beam with lateral displacement but with no rotation in the end (Figure 3a): Since the distributed load (force per length), $q(x)$, is zero, the solution to the first differential equation of Equation (8) can be expressed as:

$$E_sI\frac{d^3\varphi}{dx^3} = q(x) = 0 \;\rightarrow E_sI\varphi = \frac{C_0}{2}x^2 + C_1x + C_2$$

(10)

　　By applying the boundary condition of no rotation at the root of the cantilever beam, Equation (10) gives $C_2 = 0$. Similarly, applying the boundary condition of no rotation ($\varphi = 0$) at the end of the beam ($x = l$) to Equation (10) gives $C_1 = -\frac{C_0 l}{2}$. Substituting $C_1$ and $C_2$ in the second line of Timoshenko beam theory (Equation (8)) gives the deflection function of the beam as

$$w = \frac{C_0}{2E_sI}\left(\frac{x^3}{3} - l\frac{x^2}{2}\right) - \frac{E_sI}{\kappa AG_s}\frac{C_0}{2E_sI}(2x - l) + C_3$$

(11)

　　The two other constants ($C_0, C_3$) can be found from the boundary conditions in $x = 0$ and $x = l$. At the root ($x = 0$), the deflection equals zero, hence:

$$C_3 = -\frac{C_0 l}{2\kappa A G_s} \tag{12}$$

Moreover, the deflection at the end of the beam equals $\delta$, which gives:

$$C_0 = \frac{-\delta}{\frac{l^3}{12E_s I} + \frac{l}{\kappa A G_s}} \tag{13}$$

By substituting $C_0, C_1, C_2, C_3$ in the first line of Equation (8) and in Equation (11), the Timoshenko beam theory for a cantilever beam with lateral displacement but with no rotation could be obtained as follows:

$$\varphi = \frac{-\delta}{\frac{l^3}{6} + \frac{2E_s I l}{\kappa A G_s}}(x^2 - lx)$$

$$w = \frac{-\delta}{\frac{l^3}{12E_s I} + \frac{l}{\kappa A G_s}}\left(\frac{1}{2E_s I}\left(\frac{x^3}{3} - l\frac{x^2}{2}\right) - \frac{1}{\kappa A G_s}x\right) \tag{14}$$

Now, it is possible to find the moments and forces shown in Figure 3a. As mentioned above, the bending moment, $M_{xx}$, and the shear force, $Q_x$, are related to the angle of rotation, $\varphi$, and the deflection, $w$, by Equation (9). Therefore, by substituting $M = M_0$ and $Q_x = F$ at $x = l$ in respectively the first and second lines of Equation (9), we have:

$$M_0 = \frac{\delta}{\frac{l^2}{6E_s I} + \frac{2}{\kappa A G_s}}$$

$$F = \frac{\delta}{\frac{l^3}{12E_s I} + \frac{l}{\kappa A G_s}} \tag{15}$$

(b) A cantilever beam with rotation but with no lateral displacement in the end (Figure 3b): Since the distributed load, $q(x)$, and the boundary condition of the beam at the clamped side ($x = 0$) for this case is similar to the previous case (Case a), the relationship for angle of rotation, $\varphi$, is the same as that given in Equation (10), and also $C_2 = 0$. By considering $w = 0$ at both $x = 0$ and $x = l$, the constants $C_1$ and $C_3$ can be found as:

$$C_1 = C_0\left(\frac{2E_s I}{\kappa A G_s l} - \frac{l}{3}\right)$$

$$C_3 = \frac{C_1}{\kappa A G_s} \tag{16}$$

The beam at the free side ($x = l$) has a rotation with the known value of $\varphi = \theta$, which after being substituted in Equation (10) gives:

$$C_0 = \frac{\theta}{\frac{l^2}{6E_s I} + \frac{2}{\kappa A G_s}} \tag{17}$$

Therefore, the Timoshenko beam theory governing equations for a cantilever beam with rotation but with no lateral displacement at the end could be obtained as follows:

$$\varphi = \frac{C_0 x^2 + 2C_1 x}{2E_s I} \tag{18}$$

$$w = \frac{1}{2E_sI}\left(\frac{C_0x^3}{3} + C_1x^2\right) - \frac{C_0}{\kappa AG_s}x$$

By considering $M_{xx} = M_0$ and $Q_x = F$ at $x = l$ in Equation (9), the relationship of bending moment and force can be found as

$$M_0 = \left(\frac{\frac{2E_sI}{\kappa AG_sl} + \frac{2l}{3}}{\frac{l^2}{6E_sI} + \frac{2}{\kappa AG_s}}\right)\theta \tag{19}$$

$$F = \frac{\theta}{\frac{l^2}{6E_sI} + \frac{2}{\kappa AG_s}} \tag{20}$$

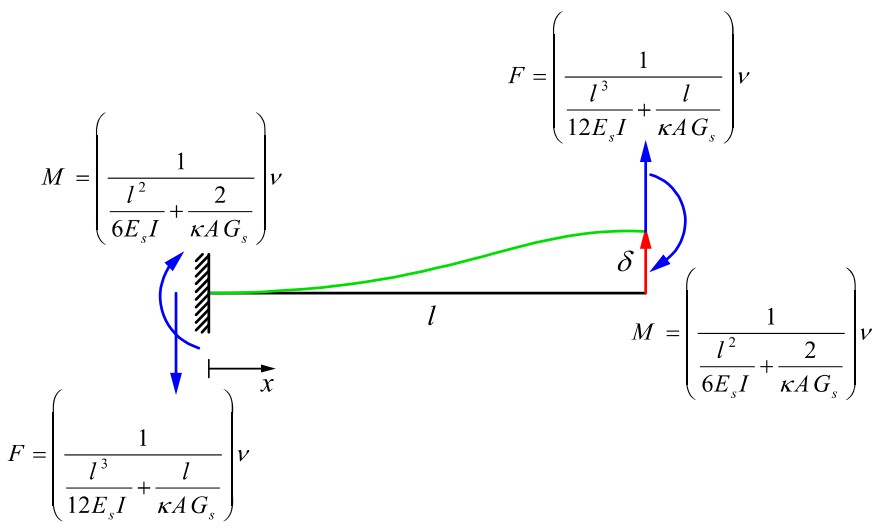

(**a**)

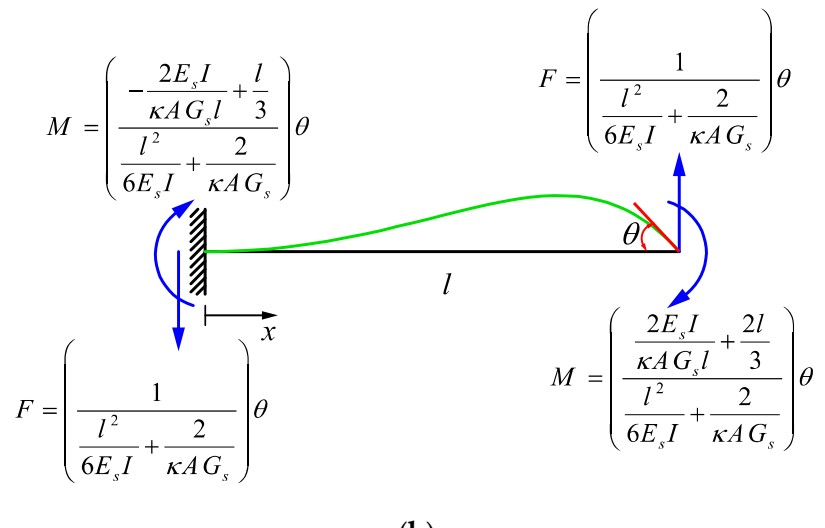

(**b**)

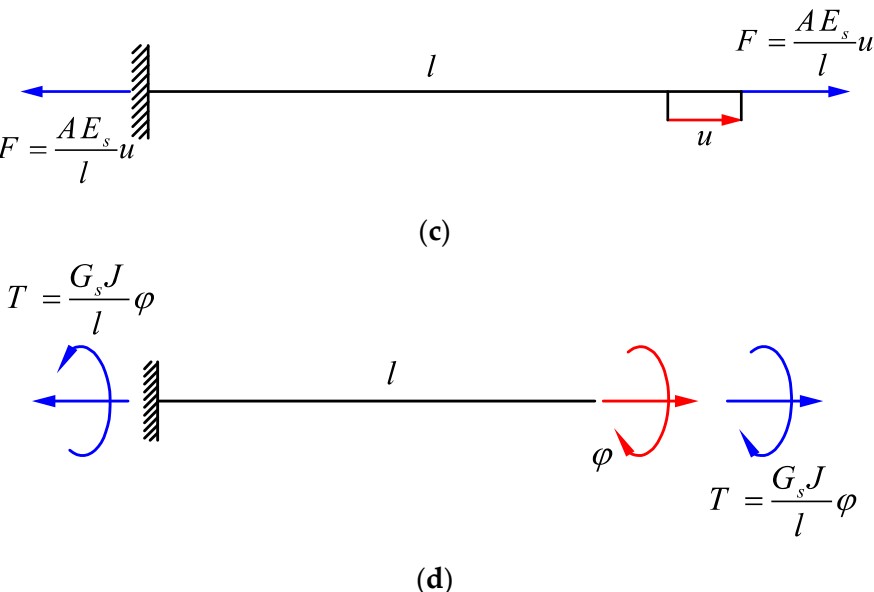

$$F = \frac{A E_s}{l} u$$

$$l$$

$$F = \frac{A E_s}{l} u$$

$$u$$

(**c**)

$$T = \frac{G_s J}{l} \varphi$$

$$l$$

$$\varphi$$

$$T = \frac{G_s J}{l} \varphi$$

(**d**)

**Figure 3.** Forces and moments required to cause (**a**) lateral displacement, $\delta$, with no rotation at the free end of the beam, (**b**) rotation, $\theta$, with no lateral displacement at the free end of the beam, (**c**) pure axial extension, and (**d**) pure twist in the free end of a Timoshenko beam.

### 2.3. From Euler–Bernoulli to Timoshenko

According to the relationships obtained for Euler–Bernoulli and Timoshenko beam theories in Sections 2.1 and 2.2, the analytical relationships for elastic modulus and Poisson's ratio of any structure based on Euler–Bernoulli theory can be converted into relationships based on Timoshenko beam theory by making the replacements suggested by Table 1 in the stiffness matrix or in the derivation formulas. To evaluate the effectiveness of the conversion approach presented in Table 1, six well known strut-based lattice structures including BCC, hexagonal packing, rhombicuboctahedron, diamond, truncated cube, and truncated octahedron (Figures 4 and 5) were considered. In the following, the procedure of converting Euler–Bernoulli formulas [26,30–32,35,39] into equivalent Timoshenko ones is described for each of the six unit cell types.

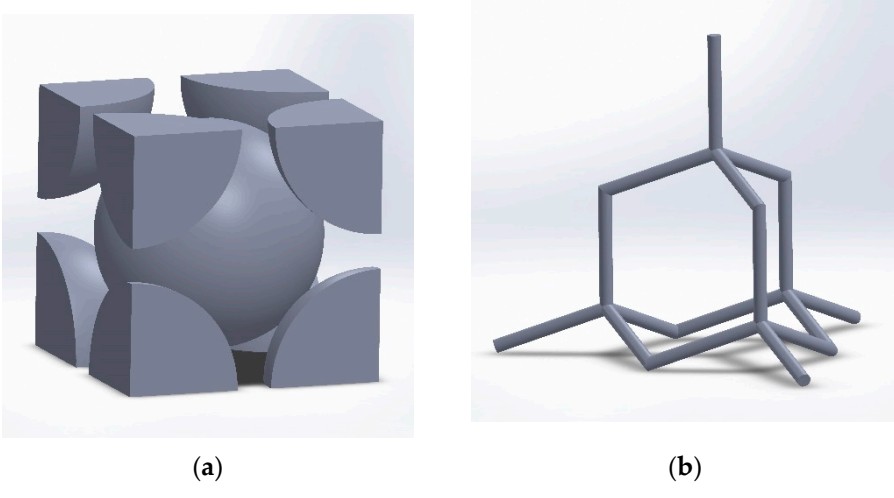

(**a**)                                    (**b**)

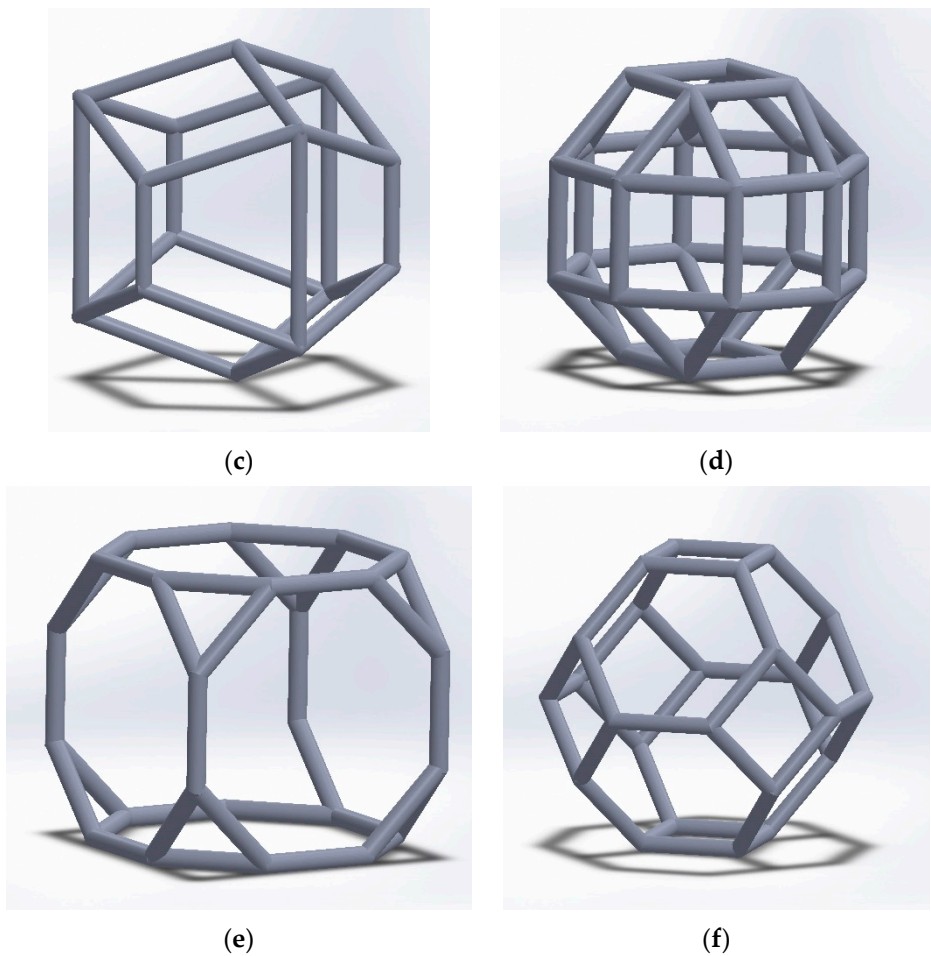

(**c**)　　　　　　　　　　　(**d**)

(**e**)　　　　　　　　　　　(**f**)

**Figure 4.** Unit cells of (**a**) body-centred cubic (BCC), (**b**) diamond, (**c**) hexagonal packing, (**d**) rhombicuboctahedron, (**e**) truncated cube, and (**f**) truncated octahedron.

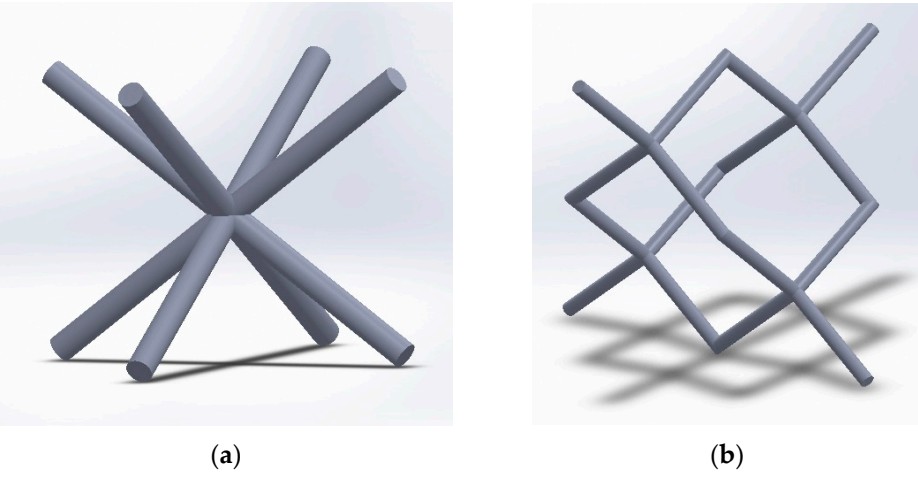

(**a**)　　　　　　　　　　　(**b**)

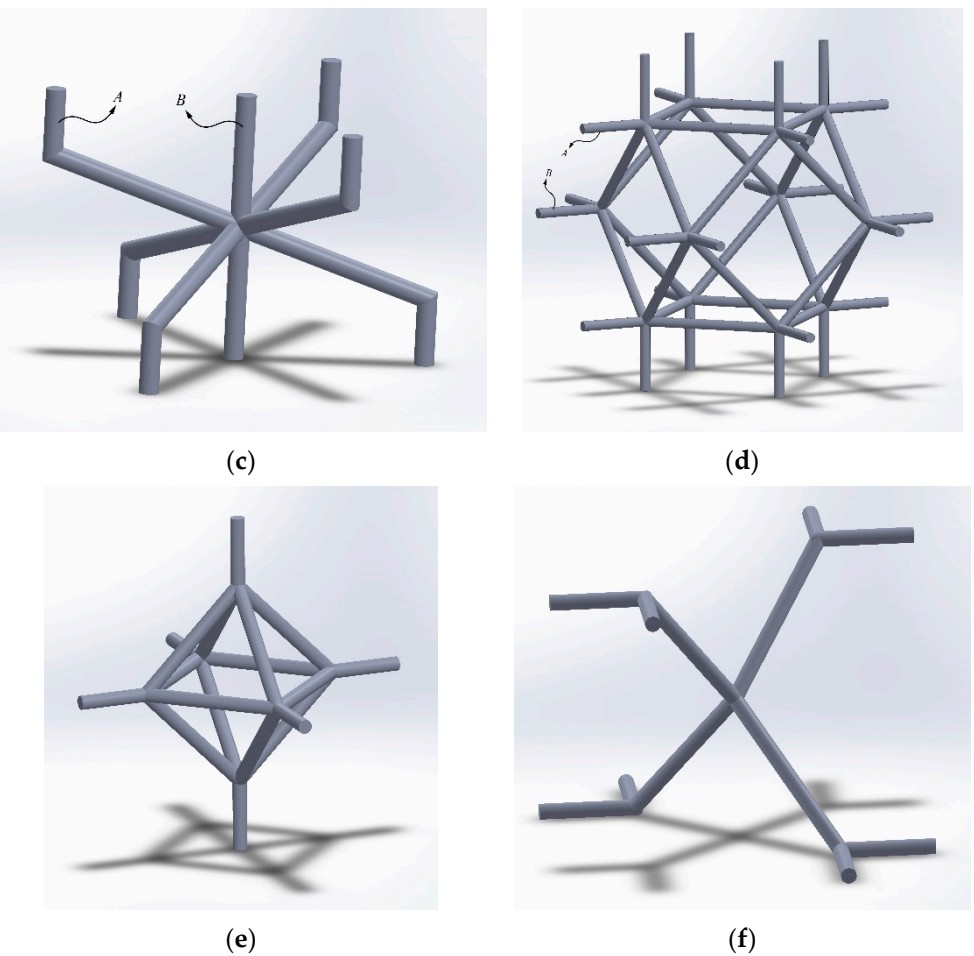

**Figure 5.** Unit cells used for analytical and numerical analysis: (**a**) BCC, (**b**) diamond, (**c**) hexagonal packing, (**d**) rhombicuboctahedron, (**e**) truncated cube, and (**f**) truncated octahedron.

**Table 1.** Conversion table for converting mechanical properties relationships based on Euler–Bernoulli beam theory to mechanical properties relationships based on Timoshenko beam theory.

| Term | Euler–Bernoulli Theory | Timoshenko Theory |
|:---:|:---:|:---:|
| Axial Tension/Compression | $\left(\dfrac{AE_s}{l}\right)u$ | $\left(\dfrac{AE_s}{l}\right)u$ |
| Torsion | $\left(\dfrac{G_sJ}{l}\right)\varphi$ | $\left(\dfrac{G_sJ}{l}\right)\varphi$ |
| Lateral deformation Force | $\left(\dfrac{12E_sI}{l^3}\right)v$ | $\left(\dfrac{1}{\dfrac{l^3}{12E_sI}+\dfrac{l}{\kappa AG_s}}\right)v$ |
| Lateral deformation Moment | $\left(\dfrac{6E_sI}{l^2}\right)v$ | $\left(\dfrac{1}{\dfrac{l^2}{6E_sI}+\dfrac{2}{\kappa AG_s}}\right)v$ |
| Rotation Force | $\left(\dfrac{6E_sI}{l^2}\right)\theta$ | $\left(\dfrac{1}{\dfrac{l^2}{6E_sI}+\dfrac{2}{\kappa AG_s}}\right)\theta$ |

| | | | |
|---|---|---|---|
| Rotation Moment | | $\left(\dfrac{4E_sI}{l}\right)\theta$ | $\left(\dfrac{\frac{2E_sI}{\kappa AG_sl}+\frac{2l}{3}}{\frac{l^2}{6E_sI}+\frac{2}{\kappa AG_s}}\right)\theta$ |

The commonly known geometries for the six noted unit cell types are presented in Figure 4, and the actual geometries used for both the analytical and numerical analyses in this study are illustrated in Figure 5. The reason of altering the unit cell shape for the BCC case is obvious as the molecular structure of BCC (Figure 4a) is composed of spheres rather than struts (Figure 5a). In the case of diamond, the unit cell shown in Figure 4b is rotated, and the unit cell in a specific direction (for which all the struts have similar angles with respect to the horizontal plane) has been considered for analysis (Figure 5b). In the other cases, the unit cell position has been shifted in order to avoid neighbouring cells having adjacent struts. If the neighbour cells have adjacent side-by-side edges, the analytical relationship obtained for the unit cell does not represent that of a lattice structure. More explanations regarding this can be found in Section 2.1.2 of [32]. Although in this paper, all the analytical relationships for the lattice structures and unit cells are presented in a normalized manner and, hence, the dimensions of the unit cells do not affect the results, an equal volume of $5 \times 5 \times 5\,mm^3$ was considered for all the unit cell types. Moreover, in each unit cell, the strut radius was increased from very small values ($r \cong 0$) to high values up to the point where relative density of the unit cell reached $\mu = 0.5$.

The conversion procedure from Euler–Bernoulli beam theory into Timoshenko beam theory for each unit cell is described in the following:

(a). BCC

The relationships for elastic modulus and Poisson's ratio of BCC unit cell have been presented in Equations (A18) and (A19) of Appendix B (extracted from [31]). In these equations, the axial extension ($\frac{AE_s}{l}$) and lateral bending ($\frac{12E_sI}{l^3}$) terms of the Euler–Bernoulli theory can be identified easily. By substituting the term $\frac{12E_sI}{l^3}$ with $\left(\dfrac{1}{\frac{l^3}{12E_sI}+\frac{l}{\kappa AG_S}}\right)$, the relationships for elastic modulus and Poisson's ratio can be converted.

(b). Diamond

For transforming relationships of elastic modulus and Poisson's ratio from Euler–Bernoulli into Timoshenko beam theory, the basic Equations (A20) and (A21) of Appendix B (extracted from [26]) for elastic modulus and (A22) and (A23) of Appendix B (extracted from [26]) for Poisson's ratio were considered. Axial extension ($\frac{AE_s}{l}$) and lateral bending ($\frac{12E_sI}{l^3}$) terms of Euler–Bernoulli theory can be identified easily. By substituting the term $\frac{12E_sI}{l^3}$ with $\left(\dfrac{1}{\frac{l^3}{12E_sI}+\frac{l}{\kappa AG_s}}\right)$, the relationships for elastic modulus and Poisson's ratio based on Timoshenko beam theory were obtained.

(c). Hexagonal packing

For this unit cell, since the analytical relationships for mechanical properties of structure have not been derived based on Euler–Bernoulli theory in the literature [35], the analytical relationships for Euler–Bernoulli and Timoshenko beam theories have both been derived in this study. The detailed derivations can be found in Appendix C.

(d). Rhombicuboctahedron

Since the final relationships for elastic modulus and Poisson's ratio of this unit cell have been presented in $\frac{r}{l}$ terms, and the stiffness matrix of unit cell contains $\frac{AE_s}{l}$, $\frac{12E_sI}{l^3}$, $\frac{G_sJ}{l}$, and $\frac{6E_sI}{l^2}$, recognition of terms provided in Table 1 in the final Equations (A24) and (A25) of Appendix B (extracted from [30]) is not possible. Therefore, by substituting the

term $\frac{12E_SI}{l^3}$ with $\left(\dfrac{1}{\frac{l^3}{12E_SI}+\frac{l}{\kappa AG_S}}\right)$ and $\frac{6E_SI}{l^2}$ with $\left(\dfrac{1}{\frac{l^2}{6E_SI}+\frac{2}{\kappa AG_S}}\right)$ in the stiffness matrix (Equation (A26) in Appendix B (which is extracted from [30]) and solving the system of equations, the relationships based on Timoshenko theory were obtained.

(e). Truncated cube

The procedure for truncated cube unit cell is very similar to what was described above for the rhombicuboctahedron unit cell. The final stiffness matrix of the unit cell in Equation (A27) of Appendix B (extracted from [32]) contains $\frac{AE_S}{l}$ and $\frac{12E_SI}{l^3}$ terms. Therefore, by substituting the term $\frac{12E_SI}{l^3}$ with $\left(\dfrac{1}{\frac{l^3}{12E_SI}+\frac{l}{\kappa AG_S}}\right)$, the stiffness matrix based on Timoshenko beam theory can be derived. Afterwards, by solving the system of equations, the mechanical properties relationships based on Timoshenko beam theory were obtained.

(f). Truncated octahedron

The relationships for elastic modulus and Poisson's ratio of truncated octahedron unit cell have been presented in Equation (A28) and (A29) of Appendix B (extracted from [39]). Since the deformation of this unit cell includes axial extension and lateral bending, the terms $\frac{AE_S}{l}$ and $\frac{12E_SI}{l^3}$ can be extracted from these equations. By substituting the term $\frac{12EI}{l^3}$ with $\left(\dfrac{1}{\frac{l^3}{12E_SI}+\frac{l}{\kappa AG_S}}\right)$, the relationships presented for elastic modulus and Poisson's ratio can be transformed into corresponding Timoshenko ones.

It is worth noting that the relationships for normalized yield stress for five of the six noted unit cells (BCC, hexagonal packing, rhombicuboctahedron, diamond, and truncated octahedron) based on Euler–Bernoulli and Timoshenko beam theories have been obtained separately in another study [40].

*2.4. Numerical Analysis*

The 3D representation of the unit cells used for numerical modelling and analysis are demonstrated in Figure 5. The actual FE models of the unit cells along with the boundary conditions are demonstrated in Appendix D. The struts of the unit cells were discretized using Timoshenko beam elements (element type BEAM189 in ANSYS), and each strut was discretized using five beam elements. The beam elements were rigidly connected to each other at their shared vertices, and they were not allowed to rotate in any direction at the connecting point. The mechanical properties of the titanium alloy Ti-6Al-4V-ELI were used for modelling the behaviour of the matrix material in the FE models. A linear elastic material model with elastic modulus and Poisson's ratio of 113.8 GPa and 0.342, respectively, was implemented. Since the BEAM189 element uses linear interpolation and takes transverse shear deformation into account, it is expected that the numerical results will be closer to the Timoshenko analytical solution. In the FE models of BCC, diamond, truncated cube, and truncated octahedron, a single unit cell with periodic boundary condition was analysed under compressive loading (Figure A2a,b,e,f in Appendix D). For the hexagonal packing and rhombicuboctahedron topologies, lattice structures consisting of 11 × 11 × 11 unit cells were used for numerical modelling. This was due to the complexity of modelling the repetitive boundary conditions in these two unit cell types. More specifically, in these two cases, each side of the unit cell is composed of two strut types (rather than one strut type in the case of other unit cell types), which have non-symmetrical displacements under compressive loading (demonstrated as strut types A and B in Figure 5c,d).

In all the FE models, the lowermost nodes of the structure were fixed in the direction parallel to the loading direction and were not allowed to rotate in any direction. For the case of FE models made out of single unit cells (Figure A2a,b,e,f in Appendix D), the side

vertices of the unit cell were constrained rotationally as they were symmetrically connected to the (imaginary) adjacent unit cells (repetitive boundary condition). In all the FE models, a downward displacement was applied on the uppermost node(s) of the structure to induce axial deformation. Moreover, the uppermost nodes were not allowed to rotate in any directions.

Mechanical properties of the FE models have been calculated based on the basic definition of elastic modulus, Poisson's ratio, and yield stress:

- Elastic modulus: The formula $E_{UC} = \frac{F_{UC}L_{UC}}{A_{UC}\delta_{UC}}$ was used for calculating numerical elastic modulus, where $L_{UC}$ is the structure length in the direction parallel to loading direction, $A_{UC}$ is the cross-sectional area of the structure in the direction perpendicular to the loading direction, $\delta_{UC}$ is the downward displacement applied to the uppermost nodes, and $F_{UC}$ is obtained by summing the reaction forces of the lowermost nodes.

- Poisson's ratio: The formula $v_{UC} = -\frac{\varepsilon_2}{\varepsilon_1} = -\frac{L_1\delta_2}{L_2\delta_1}$ was used for obtaining Poisson's ratio. In this formula, $\delta_1$ and $L_1$ are the downward displacement applied to the uppermost nodes and unit cell's length in the direction parallel to loading direction, respectively. Parameters $\delta_2$ and $L_2$ are respectively the lateral displacement of the side nodes and the structure length in the direction perpendicular to loading direction.

- Yield stress: The formula $\frac{\sigma_y}{\sigma_{ys}} = \frac{F_{UC}}{A_{UC}\,\sigma_{max}}$ was used to calculate normalized yield stress. In this formula, $\sigma_{max}$ is the maximum von Mises stress experienced in the most critical point of the structure. The critical points of each unit cell can be seen in Section 4.1.

In the cases where a lattice structure was implemented for numerical modelling (rhombicuboctahedron and hexagonal packing), all the above-mentioned terms denoted by $UC$ should rather be denoted by $Lattice$, and the lattice structure dimensions should be used for calculations.

### 3. Results

According to the initial results, by not considering the shear deformation effect in the beam theory, the forces and moments required to create a particular deformation in a single strut could be predicted by 15–20% higher for $r/l$ as large as 0.15. The complete results are presented in Appendix E.

The transformed elastic modulus, Poisson's ratio, and yields stress relationships for the six geometries are presented in Tables 2–4. For simplifying the equations, the terms $S = \frac{AE_s}{l}$, $T = \frac{1}{\frac{l^3}{12E_sI}+\frac{l}{\kappa AG_s}}$, $U = \frac{\frac{2E_sI}{\kappa AG_sl}+\frac{2l}{3}}{\frac{l^2}{6E_sI}+\frac{2}{\kappa AG_s}}$, $V = \frac{1}{\frac{l^2}{6E_sI}+\frac{2}{\kappa AG_s}}$, and $W = \frac{G_sJ}{l}$ and have been used in the noted tables for the relationships based on the Timoshenko beam theory. For four of the geometries (BCC [31], rhombicuboctahedron [30], truncated cube [32], and truncated octahedron [39]), the original Euler–Bernoulli relationships have been presented in Tables 2 and 3 as well. In the case of hexagonal packing, the original Euler–Bernoulli relationships [35] for elastic modulus and Poisson's ratio have been improved and adjusted (the description on how and why can be found in Appendix C), and the improved relationships are presented in Tables 2 and 3. For the case of diamond unit cell, the original rela-

tionships for elastic modulus and Poisson's ratio have been conserved but stated as a function of $r/l$ rather than of relative density, $\mu$. As for the yield stress (Table 4), the analytical Euler–Bernoulli relationships presented in our other paper [40] have been presented.

In order to compare the already-existing [26,30–32,35,39] or improved Euler–Bernoulli analytical relationships with the newly transformed Timoshenko analytical relationships, the results of the analytical relationships for both the Euler–Bernoulli and Timoshenko beam theory have been compared with their numerical and experimental counterparts in Figures 6–8. The relative density relationships for the six geometries are presented in Table 5.

For all the geometries, the newly transformed Timoshenko relationships have exceptionally good agreement with the numerical results for all the mechanical properties: elastic modulus, Poisson's ratio, and yield stress (Figures 6–8). As for the previously obtained formulas obtained in the literature or newly adjusted Euler–Bernoulli formulas, the maximum difference between Euler–Bernoulli analytical elastic modulus and corresponding numerical values for BCC, diamond, hexagonal packing, rhombicuboctahedron, truncated cube, and truncated octahedron (at relative density of $\mu = 0.5$) are, respectively, 21.34%, 57.71%, 20.21%, 14.52%, 14.98%, and 45.54%. However, the corresponding differences between the Timoshenko analytical relationships and the numerical values for the noted geometries are, respectively, 1.13%, 2.21%, 8.29%, 2.97%, 0.43%, and 3.15%.

**Table 2.** Normalized elastic modulus relationships based on Euler–Bernoulli and Timoshenko beam theories for different unit cell types ($S = \frac{AE_s}{l}$, $T = \frac{1}{\frac{l^3}{12E_sI}+\frac{l}{\kappa AG_s}}$, $V = \frac{1}{\frac{l^2}{6E_sI}+\frac{2}{\kappa AG_s}}$, and $W = \frac{G_sJ}{l}$).

| Unit Cell | Relative Elastic Modulus, $E/E_s$ | |
| --- | --- | --- |
| | **Euler–Bernoulli Theory** | **Timoshenko Theory** |
| BCC | $\dfrac{4\sqrt{3}}{\left(\dfrac{l^2}{\pi r^2}+\dfrac{l^4}{2\pi r^4}\right)}$ <br> [31] | $\dfrac{4\sqrt{3}}{E\left(\dfrac{4l}{3S}+\dfrac{8l}{3T}\right)}$ |
| Diamond | $\dfrac{3\sqrt{3}}{\dfrac{8}{3\pi}\left(\dfrac{l}{r}\right)^4+\dfrac{4}{\pi}\left(\dfrac{l}{r}\right)^2}$ <br> [26] | $\dfrac{3\sqrt{3}}{E\left(\dfrac{8l}{T}+\dfrac{4l}{S}\right)}$ |
| Hexagonal packing | $\dfrac{\pi\sqrt{3}}{4}\left(\dfrac{r}{l}\right)^2\left(1+\dfrac{1}{\dfrac{5}{9}+\dfrac{4}{27}\left(\dfrac{l}{r}\right)^2}\right)$ <br> (see Appendix C) | $\dfrac{\sqrt{3}S\left(1+\dfrac{1}{\dfrac{5}{9}+\dfrac{4S}{9T}}\right)}{4El}$ |
| Rhombicuboctahedron | $\dfrac{4\pi\left(\frac{r}{l}\right)^2}{3(1+\sqrt{2})}\left[\dfrac{4+108\left(\frac{r}{l}\right)^2+207\left(\frac{r}{l}\right)^4+81\left(\frac{r}{l}\right)^6+\frac{G}{E}\left(\frac{2}{3}+19\left(\frac{r}{l}\right)^2+45\left(\frac{r}{l}\right)^4+18\left(\frac{r}{l}\right)^6\right)}{8+70\left(\frac{r}{l}\right)^2+105\left(\frac{r}{l}\right)^4+27\left(\frac{r}{l}\right)^6+\frac{G}{E}\left(\frac{4}{3}+13\left(\frac{r}{l}\right)^2+23\left(\frac{r}{l}\right)^4+6\left(\frac{r}{l}\right)^6\right)}\right]$ <br> [30] | $\dfrac{4S(2WS^3+19WS^2T+15WST^2+2WT^3+4S^3V+36S^2TV+23ST^2V+3T^3V)\mathrm{L}_{UC}}{E(12WS^3+39WS^2T+23WST^2+2WT^3+24S^3V+70S^2TV+35ST^2V+3T^3V)\mathrm{A}_{UC}}$ |
| Truncated cube | $\dfrac{2\pi}{\sqrt{2}+1}\left(\dfrac{r}{l}\right)^2\dfrac{1+9\left(\frac{r}{l}\right)^2}{5+21\left(\frac{r}{l}\right)^2}$ <br> [32] | $\dfrac{8S(S+3T)\mathrm{L}_{UC}}{E(5S+7T)\mathrm{A}_{UC}}$ |
| Truncated octahedron | $\dfrac{6\sqrt{2}I}{l^4\left(1+\dfrac{12I}{Al^2}\right)}$ <br> [39] | $\dfrac{1}{\sqrt{2}El\left(\dfrac{1}{S}+\dfrac{1}{T}\right)}$ |

**Table 3.** Poisson's ratio relationships based on Euler–Bernoulli and Timoshenko beam theories for different unit cell types ($S = \frac{AE_s}{l}$, $T = \frac{1}{\frac{l^3}{12E_sI}+\frac{l}{\kappa AG_s}}$, $V = \frac{1}{\frac{l^2}{6E_sI}+\frac{2}{\kappa AG_s}}$, $W = \frac{G_sJ}{l}$).

| Unit Cell | Poisson's Ratio, $v$ | |
| --- | --- | --- |
| | **Euler–Bernoulli Theory** | **Timoshenko Theory** |
| BCC | $\dfrac{-\dfrac{1}{\pi r^2}+\dfrac{l^2}{4\pi r^4}}{\dfrac{1}{\pi r^2}+\dfrac{l^2}{2\pi r^4}}$ <br> [31] | $\dfrac{-\dfrac{1}{S}+\dfrac{1}{T}}{\dfrac{1}{S}+\dfrac{2}{T}}$ |
| Diamond | $\dfrac{1-3\left(\dfrac{r}{l}\right)^2}{2-3\left(\dfrac{r}{l}\right)^2}$ <br> [26] | $\dfrac{-\dfrac{1}{S}+\dfrac{1}{T}}{\dfrac{1}{S}+\dfrac{2}{T}}$ |
| Hexagonal packing | $\dfrac{-\dfrac{1}{9}+\dfrac{1}{27}\left(\dfrac{l}{r}\right)^2}{\dfrac{5}{9}+\dfrac{4}{27}\left(\dfrac{l}{r}\right)^2}$ <br> (see Appendix C) | $\dfrac{\dfrac{S}{T}-1}{5+\dfrac{4S}{T}}$ |
| Rhombicuboctahedron | $\dfrac{1}{3}\left[\dfrac{8-12\left(\dfrac{r}{l}\right)^2-36\left(\dfrac{r}{l}\right)^4+\dfrac{G}{E}\left(\dfrac{4}{3}-\left(\dfrac{r}{l}\right)^2-9\left(\dfrac{r}{l}\right)^4\right)}{8+70\left(\dfrac{r}{l}\right)^2+105\left(\dfrac{r}{l}\right)^4+27\left(\dfrac{r}{l}\right)^6+\dfrac{G}{E}\left(\dfrac{4}{3}+13\left(\dfrac{r}{l}\right)^2+23\left(\dfrac{r}{l}\right)^4+6\left(\dfrac{r}{l}\right)^6\right)}\right]$ <br> [30] | $\dfrac{S(S-T)(4WS+3WT+8SV+4TV)}{12WS^3+39WS^2T+23WST^2+2WT^3+24S^3V+70S^2TV+35ST^2V+3T^3V}$ |
| Truncated cube | $\dfrac{1-3\left(\dfrac{r}{l}\right)^2}{5+21\left(\dfrac{r}{l}\right)^2}$ <br> [32] | $\dfrac{\dfrac{1}{T}-\dfrac{1}{S}}{\dfrac{5}{T}+\dfrac{7}{S}}$ |
| Truncated octahedron | $0.5\dfrac{Al^2-12I}{Al^2+12I}$ <br> [39] | $\dfrac{1}{2}\dfrac{S-T}{S+T}$ |

**Table 4.** Relative yield stress relationships based on Euler–Bernoulli and Timoshenko beam theories for different unit cell types ($S = \frac{AE_s}{l}$, $= \frac{1}{\frac{l^3}{12E_sI}+\frac{l}{\kappa AG_s}}$, $U = \frac{\frac{2E_sI}{\kappa AGl}+\frac{2l}{3}}{\frac{l^2}{6E_sI}+\frac{2}{\kappa AG_s}}$, $V = \frac{1}{\frac{l^2}{6E_sI}+\frac{2}{\kappa AG_s}}$, and $W = \frac{G_sJ}{l}$).

| Unit Cell | Relative Yield Stress, $\frac{\sigma_y}{\sigma_{ys}}$ | |
| :---: | :---: | :---: |
| | **Euler–Bernoulli Theory** | **Timoshenko Theory** |
| BCC | $\dfrac{1}{\frac{1}{3\pi\sqrt{3}}\left(\frac{l}{r}\right)^2+\frac{4}{\pi\sqrt{6}}\left(\frac{l}{r}\right)^3}$ <br> [40] | $\dfrac{3\sqrt{3}}{l^2\left(\frac{1}{A}+\frac{cl}{\sqrt{2}I}\right)}$ |
| Diamond | $\dfrac{1}{\frac{4}{3\pi\sqrt{3}}\left(\frac{l}{r}\right)^2+\frac{8\sqrt{2}}{\pi\sqrt{3}}\left(\frac{l}{r}\right)^3}$ <br> [40] | $\dfrac{3\sqrt{3}}{4l^2\left(\frac{1}{A}+\frac{cl}{\sqrt{2}I}\right)}$ |
| Hexagonal packing | $\dfrac{\pi\sqrt{3}}{4}\left(\frac{r}{l}\right)^2\left(1+\dfrac{1}{\frac{5}{9}+\frac{4}{27}\left(\frac{l}{r}\right)^2}\right)$ <br> [40] | $\dfrac{\sqrt{3}S\left(1+\dfrac{1}{\frac{5}{9}+\frac{4S}{9T}}\right)}{4El}$ |
| Rhombicuboctahedron | $\dfrac{4A}{A_{UC}}\left(1-\dfrac{6\sqrt{2}(A^2l^4-6AIl^2-72I^2)rAl}{1728\left(4.5+\frac{G_s}{E_s}\right)I^3+1080\left(4.6+\frac{G_s}{E_s}\right)AI^2l^2+6\left(108+19\frac{G_s}{E_s}\right)A^2Il^4+\left(6+\frac{G_s}{E_s}\right)A^3l^6+6\sqrt{2}(A^2l^4-6AIl^2-72I^2)rAl}\right)$ | * (See the footnote of the table) |
| Truncated cube | $\dfrac{\pi}{\left(\sqrt{2}+1\right)^2}\left(\frac{r}{l}\right)^2$ <br> [4] | $\dfrac{A}{l^2\left(\sqrt{2}+1\right)^2}$ |
| Truncated octahedron | $\dfrac{1}{\frac{\sqrt{2}}{2\pi}\left(\frac{l}{r}\right)^2+\frac{\sqrt{2}}{\pi}\left(\frac{l}{r}\right)^3}$ <br> [40] | $\dfrac{\sqrt{2}}{l^2\left(\dfrac{Vc\left(\frac{3}{T}+\frac{1}{S}\right)}{I}+\dfrac{S\left(\frac{1}{T}+\frac{3}{S}\right)}{A}\right)}$ |

\* The Timoshenko-based relationship for the yield stress of rhombicuboctahedron is:

$$\frac{16IA}{A_{UC}}\left(\frac{2\acute{W}(\acute{S}^3+\acute{T}^3)+19\acute{W}\acute{S}^2\acute{T}+15\acute{W}\acute{S}\acute{T}^2+4\acute{S}^3\acute{V}+36\acute{S}^2\acute{T}\acute{V}+23\acute{T}^2\acute{V}+3\acute{T}^3\acute{V}}{2I(2\acute{W}\acute{S}^3+2\acute{W}\acute{T}^3+19\acute{W}\acute{S}^2\acute{T}+15\acute{W}\acute{S}\acute{T}^2+4\acute{S}^3\acute{V}+36\acute{S}^2\acute{T}\acute{V}+23\acute{S}\acute{T}^2\acute{V}+3\acute{T}^3\acute{V})+\sqrt{2}rA(8\acute{S}^2\acute{U}\acute{V}-4\acute{S}\acute{T}\acute{U}\acute{V}-4\acute{T}^2\acute{U}\acute{V}-2\acute{S}^2\acute{V}^2+\acute{S}\acute{T}\acute{V}^2+\acute{T}^2\acute{V}^2)}\right)$$

where, $\acute{S}=\frac{2AE_s}{l}$, $\acute{T}=\frac{1}{\frac{l^3}{96E_sI}+\frac{l}{\kappa AG_s}}$, $\acute{U}=\frac{\frac{4E_sI}{\kappa AG_sl}+\frac{l}{3}}{\frac{l^2}{24E_sI}+\frac{2}{\kappa AG_s}}$, $\acute{V}=\frac{1}{\frac{l^2}{24E_sI}+\frac{2}{\kappa AG_s}}$, $\acute{W}=\frac{2G_sJ}{l}$.

**Table 5.** Relative density relationships for different unit cell types.

| Unit Cell | $\mu$, Relative Density |
|---|---|
| BCC | $3\sqrt{3}\pi\left(\dfrac{r}{l}\right)^2 - 4\sqrt{6}\pi\left(\dfrac{r}{l}\right)^3$ <br> [40] |
| Diamond | $\dfrac{3\sqrt{3}\pi}{4}\left(\dfrac{r}{l}\right)^2 - \dfrac{9\sqrt{2}}{4}\left(\dfrac{r}{l}\right)^3$ |
| Hexagonal packing | $\dfrac{5\pi}{2\sqrt{3}}\left(\dfrac{r}{l}\right)^2$ |
| Rhombicuboctahedron | $\dfrac{36\pi}{7+\sqrt{5}}\left(\dfrac{r}{l}\right)^2 - \dfrac{12(12.0404)}{7+\sqrt{5}}\left(\dfrac{r}{l}\right)^3$ <br> [30] |
| Truncated cube | $\dfrac{15\pi}{\left(1+\sqrt{3}\right)^3}\left(\dfrac{r}{l}\right)^2$ <br> [32] |
| Truncated octahedron | $\dfrac{3\pi}{2\sqrt{2}}\left(\dfrac{r}{l}\right)^2$ <br> [39] |

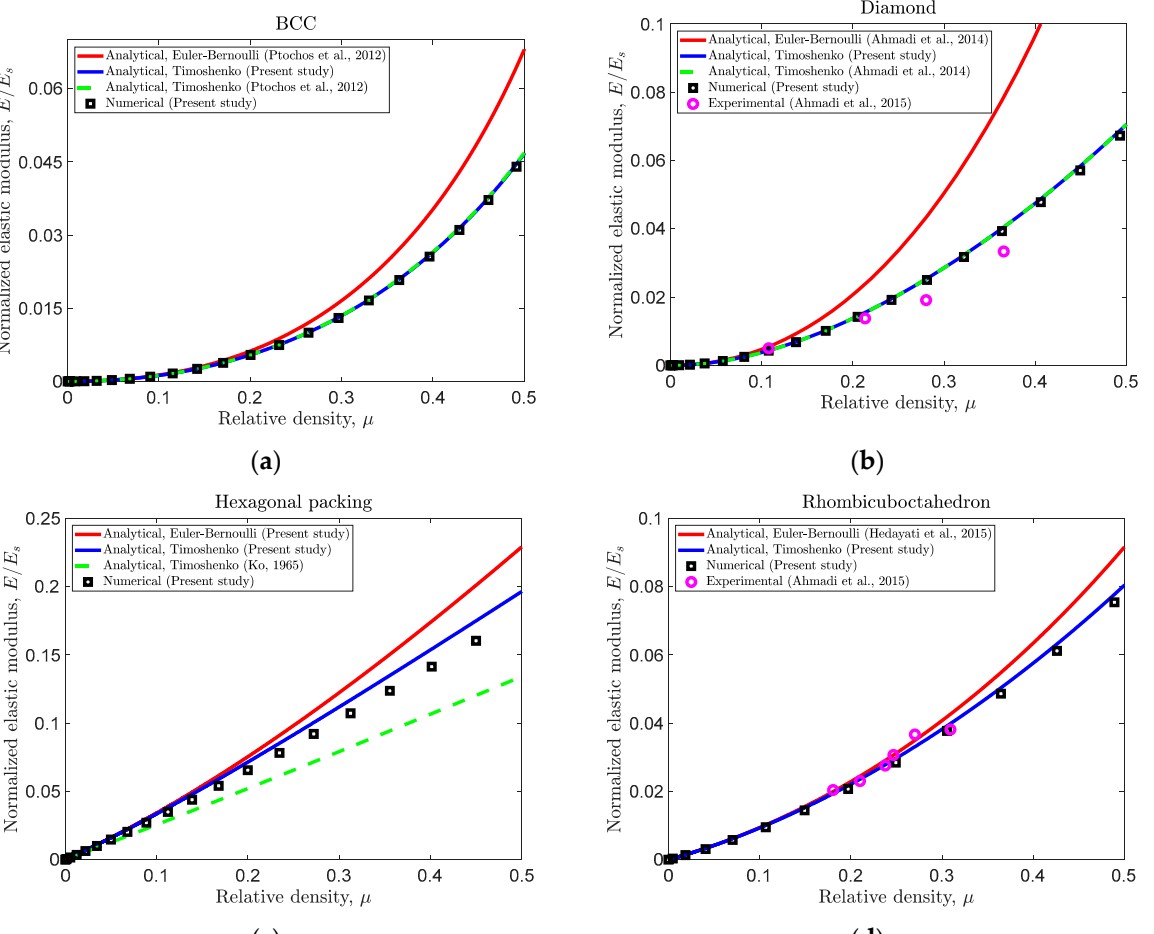

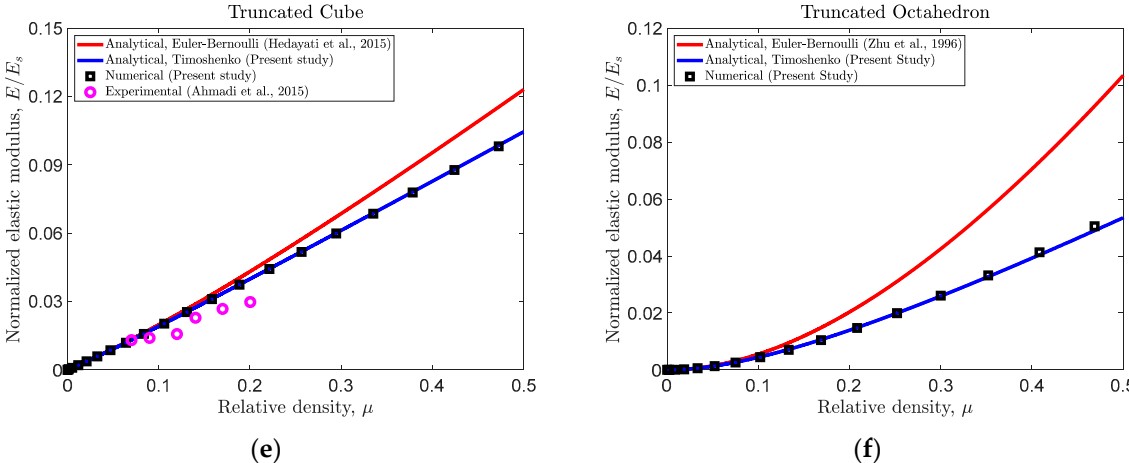

(**e**)                    (**f**)

**Figure 6.** Comparison of analytical (Euler–Bernoulli and Timoshenko) and numerical values of normalized elastic modulus for different unit cell types: (**a**) BCC, (**b**) diamond, (**c**) hexagonal packing, (**d**) rhombicuboctahedron, (**e**) truncated cube, and (**f**) truncated octahedron.

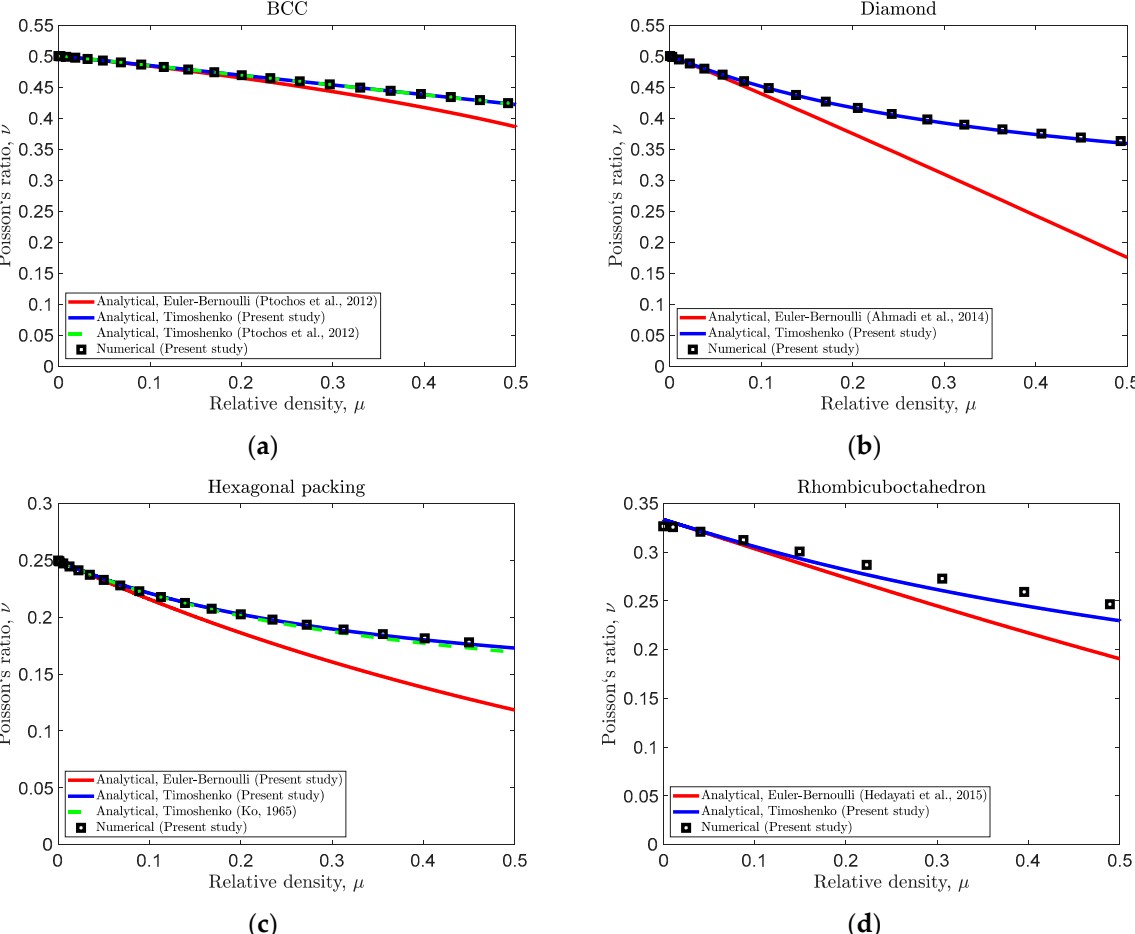

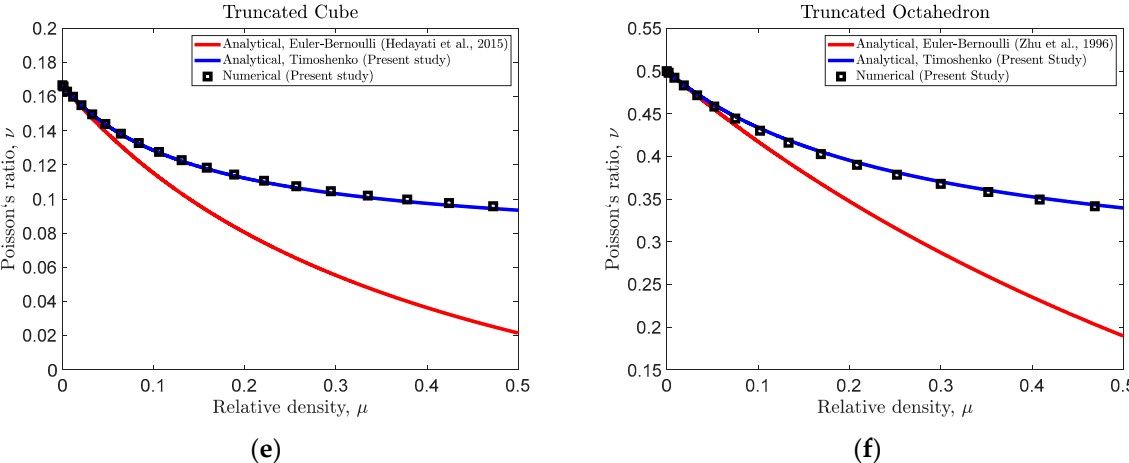

**Figure 7.** Comparison of analytical (Euler–Bernoulli and Timoshenko) and numerical values of Poisson's ratio for different unit cell types: (**a**) BCC, (**b**) diamond, (**c**) hexagonal packing, (**d**) rhombicuboctahedron, (**e**) truncated cube, and (**f**) truncated octahedron.

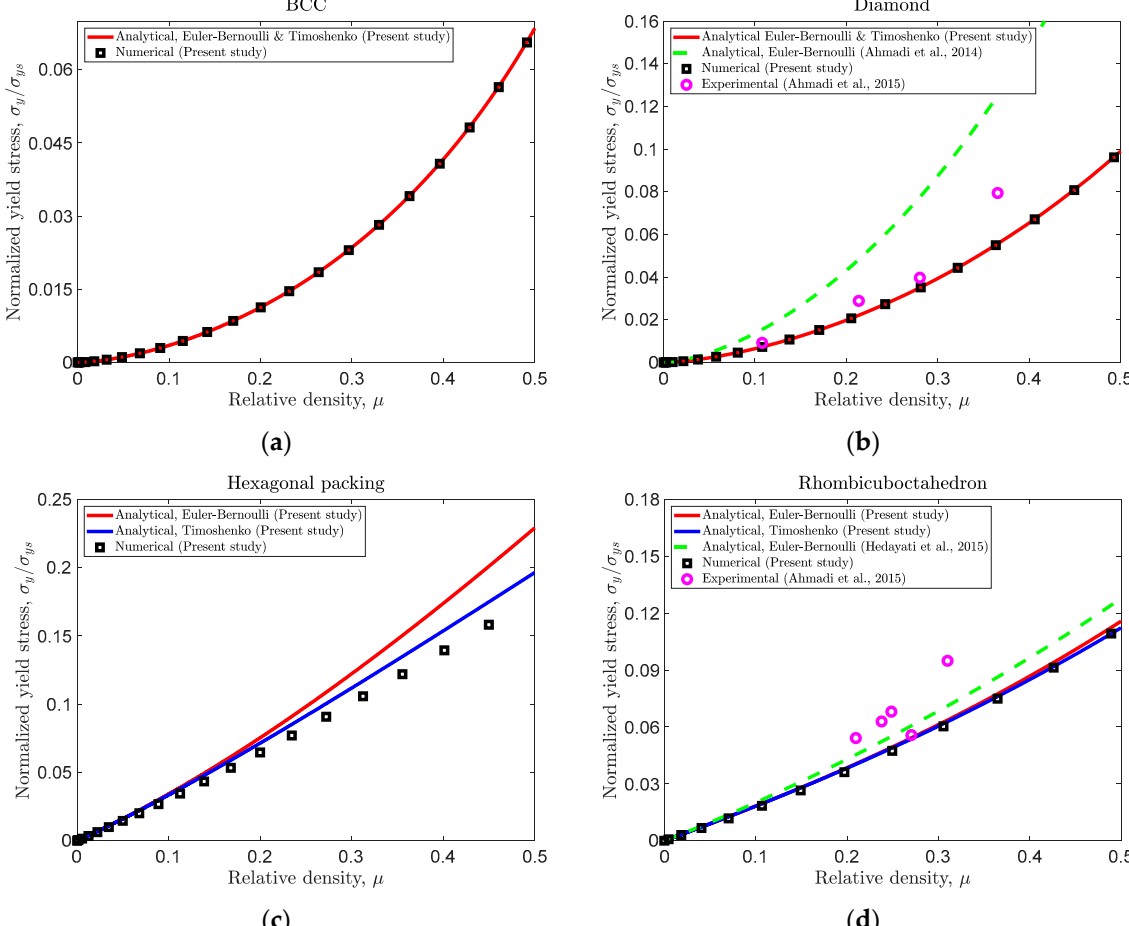

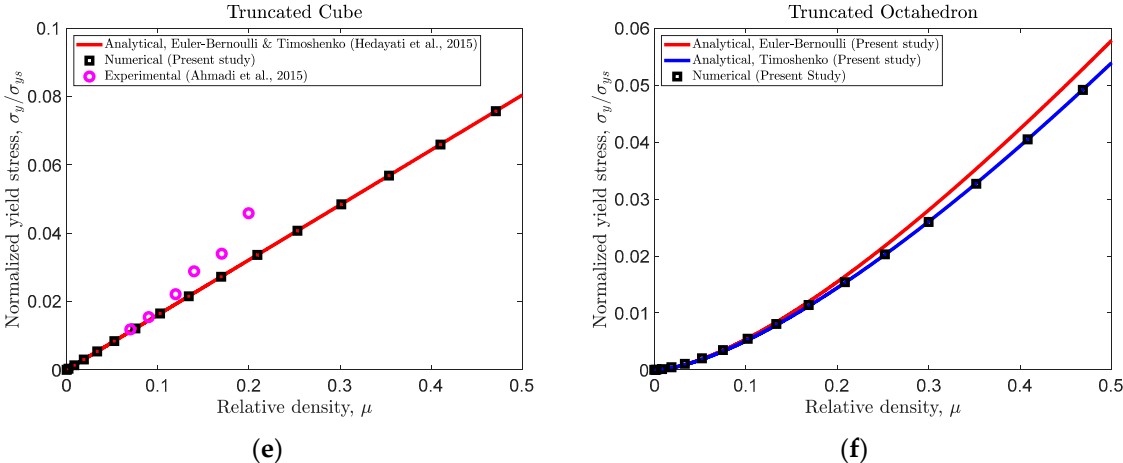

**Figure 8.** Comparison of analytical (Euler–Bernoulli and Timoshenko) and numerical values of yield stress for different unit cell types: (**a**) BCC, (**b**) diamond, (**c**) hexagonal packing, (**d**) rhombicuboctahedron, (**e**) truncated cube, and (**f**) truncated octahedron.

Similarly, the maximum differences between Euler–Bernoulli analytical Poisson's ratio and corresponding numerical values for BCC, diamond, hexagonal packing, rhombicuboctahedron, truncated cube, and truncated octahedron are, respectively, 3.07%, 27.69%, 28.00%, 21.54%, 73.58%, and 40.51%. The corresponding differences between the Timoshenko analytical relationships and the numerical values for the noted geometries are, respectively, 0.133%, 0.826%, 0.899%, 6.24%, 1.36%, and 0.498%, which shows a significant improvement in the accuracy of the analytical relationships. As can be seen in Figures 6–8, the numerical/analytical discrepancy for the case of Timoshenko beam theory is significantly less (around 1/10) than that of Euler–Bernoulli beam theory.

As for the yield stress (Figure 8), the analytical Euler–Bernoulli and Timoshenko relationships are identical for BCC, diamond, and truncated cube unit cells. The reason is explained in Section 4.3. However, for three other geometries, namely hexagonal packing, rhombicuboctahedron, and truncated octahedron, the analytical relationships based on Euler–Bernoulli differ from the relationships obtained based on Timoshenko beam theory. Nevertheless, for all the cases, the Timoshenko analytical yield stress curve has exceptionally good agreement with numerical results (Figure 8). For BCC, diamond, and truncated cube unit cells (the geometries that have the same yield stress analytical relationships for Euler–Bernoulli and Timoshenko beam theories), the maximum difference between analytical and numerical values is less than 0.25%. As for the three other geometries, the maximum difference between Euler–Bernoulli analytical normalized yield stress and corresponding numerical values for hexagonal packing, rhombicuboctahedron and truncated octahedron are, respectively, 21.25%, 3.02%, and 7.52%. However, such differences for the analytical relationships based on Timoshenko beam theory and numerical values are, respectively, 9.49%, 0.09%, and 0.162%, which shows a significant improvement in the accuracy of the analytical relationships.

As for the proximity of the analytical/numerical elastic modulus and yield stress values to the experimental data, the experimental data points have been presented in Figures 6 and 8 for rhombicuboctahedron, truncated cube, and diamond unit cells only, as in the literature, there are experimental measurements for such unit cells only [41]. As for the elastic modulus, converting the analytical relationships from Euler–Bernoulli to Timoshenko has led to closer proximity of analytical and experimental values (Figure 6b,d,e). As for the yield stress, for the three geometries for which experimental data points are available (diamond, rhombicuboctahedron, and truncated cube), the analytical relationships based on Euler–Bernoulli and Timoshenko are identical or almost overlapping, and both are in good agreement with experimental data points (Figure 8b,d,e).

## 4. Discussions

### 4.1. Unit Cell's Behaviour

The main mechanism of deformation in the struts of lattice structures and porous materials is flexure, stretching/contraction, or a combination of them. In the unit cells with a high fraction of vertical struts (struts aligned with the loading condition), the main reason for collapse is the axial normal stresses generated in their struts. Therefore, the deformation in these structures is stretching-dominated and the lattice structure collapses due to the generation of unbearable axial stress in the struts. As can be seen in Figure 9, the hexagonal packing and truncated cube unit cells could be considered as stretching-dominated structures due to the high presence of vertical struts in their architecture, while the other structures namely BCC, diamond, truncated octahedron as well as rhombicuboctahedron could be considered as bending-dominated structure, as they are mostly made up of oblique struts. The main mode of deformation in a unit cell (stretching-dominated or bending-dominated) determines the general deformation of the unit cell and, therefore, its stiffness and yield strength. According to Figure 9, the critical points of the hexagonal packing and truncated cube unit cells are located in the vertical struts due to their stretching-dominated behaviour. On the other hand, the critical points of BCC, diamond, truncated octahedron, and rhombicuboctahedron unit cells are located at the end of oblique struts due to their bending-dominated behaviour. In addition, it can be seen in Figures 6 and 8 that among all the unit cell types, the hexagonal packing and truncated cube structures have the highest stiffness and yield strength, especially in lower values of relative density due to their stretching-dominated behaviour. It is worth noting that since the rhombicuboctahedron has vertical struts, it has an in-between behaviour and gives higher stiffness and strength in comparison with other bending-dominated unit cells. More figures that illustrate the axial and bending stresses in the struts of the unit cells can be found in Appendix D.

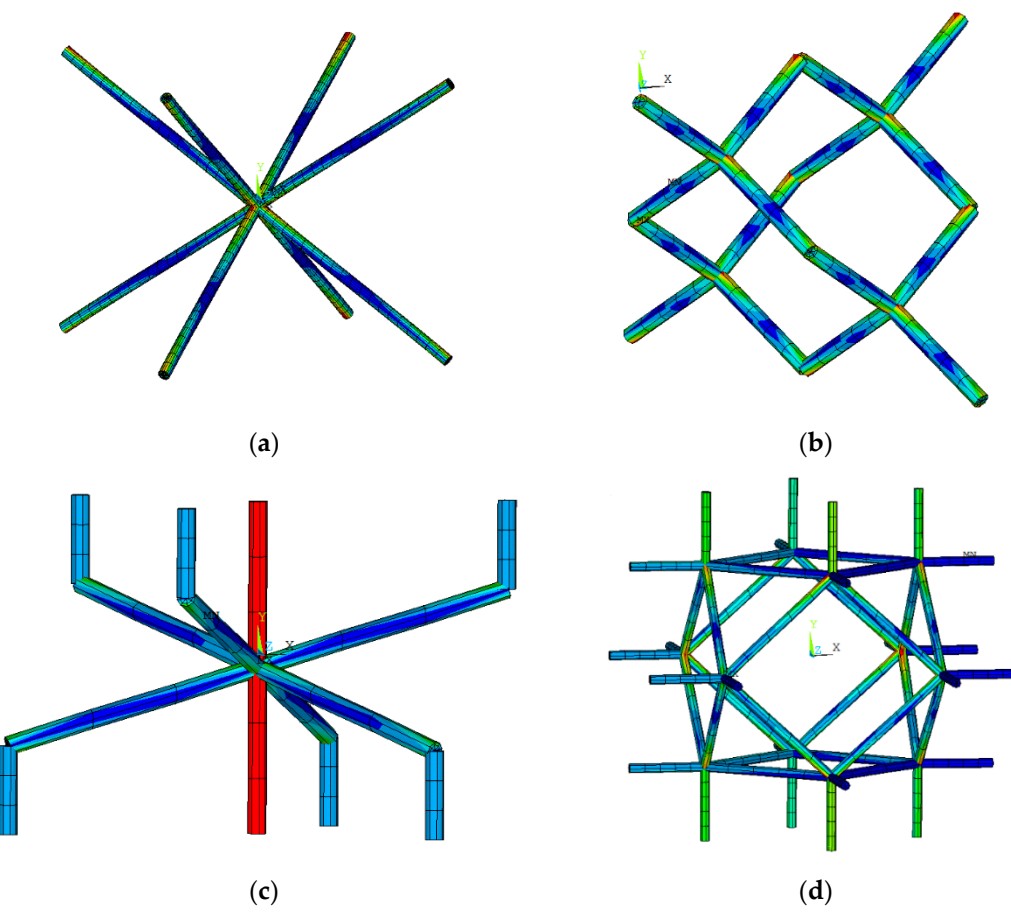

(a)　　　　　　　　　　　　　　　　　　　　　　(b)

(c)　　　　　　　　　　　　　　　　　　　　　　(d)

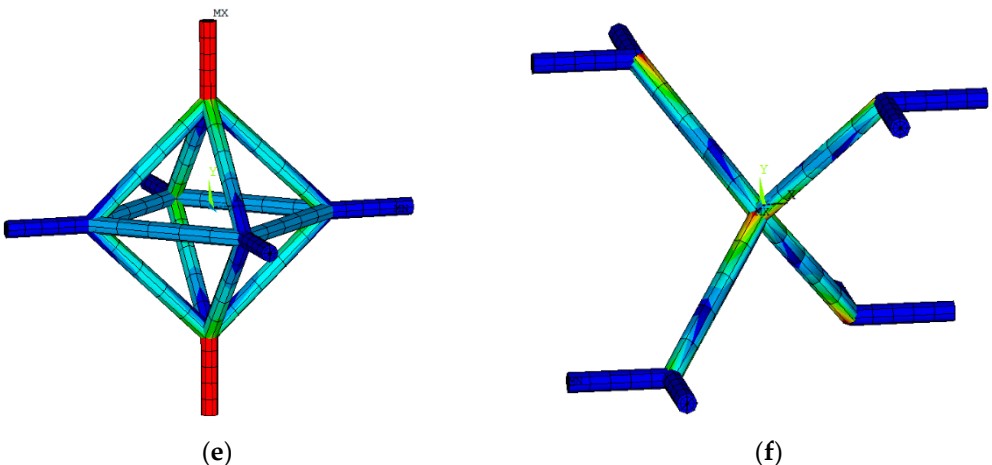

<div align="center">(<b>e</b>)                     (<b>f</b>)</div>

**Figure 9.** Unit cells stress contour for (**a**) BCC, (**b**) diamond, (**c**) hexagonal packing, (**d**) rhombicuboctahedron, (**e**) truncated cube, and (**f**) truncated octahedron unit cells.

### 4.2. Why the New Relationships Give Much Better Accuracy?

In this paper, new relationships based on Timoshenko beam theory have been derived for elastic modulus, Poisson's ratio, and yield strength of several topologies. In addition to implementing Timoshenko beam theory, some adjustments in deriving analytical relationships based on Euler–Bernoulli beam theory have been implemented for hexagonal packing unit cell (see Appendix C). Timoshenko beam theory takes into account shear deformation and rotational bending effects, making it suitable for describing the behaviour of thick beams. As a result, the new relationships based on Timoshenko beam theory give much better accuracy even at high relative densities. This improvement is significant for analytical/experimental agreement and exceptional for analytical/numerical agreement. To give a more physically tangible understanding, it must be noted that taking into account the shear deformation effect increases the flexibility of the beam, which effectively leads to larger deflections of the struts (and therefore the whole lattice structure) under an imposed load. This leads to respectively lower and higher elastic modulus and Poisson's ratio of the structure for Timoshenko theory as compared to Euler–Bernoulli theory. It is worth noting that in the bending-dominated unit cells, the discrepancy between the Euler–Bernoulli and Timoshenko results is much greater as compared to that for stretch-dominated unit cells (Figure 6). In other words, the inability of the Euler–Bernoulli beam theory to predict the effective elastic moduli of open-cell structures based on BCC, diamond, and truncated octahedron unit cells can be attributed to their main mode of deformation (bending-domination) and the importance of considering the shear deformation effect.

### 4.3. Yield Strength

As mentioned in the Results section, the normalized yield stress relationships based on Timoshenko and Euler–Bernoulli theories are identical for BCC, diamond, and truncated cube structures and both theories have good agreement with the numerical results. However, for the hexagonal packing, truncated octahedron, and rhombicuboctahedron structures, the Euler–Bernoulli and Timoshenko yield strength results are quite different (particularly for the hexagonal packing case), and the Timoshenko analytical results show much better overlapping with numerical results as compared to that for the Euler–Bernoulli analytical curve. The reason why the Timoshenko and Euler–Bernoulli yield strengths are identical for some geometries and different for some other is described extensively in [40], but it is explained here briefly. As introduced above, $T = \frac{12 E_s I}{l^3}$ and $V =$

$\frac{6E_sI}{l^2}$ for Euler–Bernoulli beam theory, and $T = \frac{1}{\frac{l^3}{12E_sI} + \frac{l}{\kappa A G_s}}$ and $V = \frac{1}{\frac{l^2}{6E_sI} + \frac{2}{\kappa A G_s}}$ for Timoshenko beam theory. For both the theories, $\frac{V}{T} = \frac{l}{2}$. Therefore, as the analytical yield strength relationship for both the diamond and BCC in the final stage of derivation is a mere function of $\frac{V}{T}$, both the theories give identical relationships in these two unit cells. As for the truncated cube case, the normalized yield stress in the critical strut could be obtained from the equilibrium of forces and moments, and the failure is caused merely due to axial stress. Therefore, the Euler–Bernoulli and Timoshenko theories give the same results again. Nevertheless, for the three other unit cells, first, the displacement caused by bending needs to be taken into account in the calculations, and second, the formulas in the final stages of derivation are not a mere function of $\frac{V}{T}$, and the V and T terms are present independently and not in a fraction form in such a way that one is a factor of the other. Therefore, the results differ for Euler–Bernoulli and Timoshenko theories for the three other unit cells.

### 4.4. Some Points Regarding Experimental Data Points

Both analytical and numerical techniques over-predict the experimental elastic modulus of the diamond and truncated cube structures, while they under-predict their experimental yield stress. The random irregularities and imperfections created during the AM process diminish the elastic modulus of the lattice structures. These defects create weak links in the structure that lower the mechanical properties of structures. However, in the case of yield stress, when the initial regions of the lattice structures in the critical points (which experience the highest stress levels in the whole lattice structure) are yielded, their local yield stress increases due to strain-hardening phenomenon, and the structure is still able to remain almost intact as the strain-hardening strengthens the structure at the initial damage points. However, when the external load is increased to higher values, the second (and possibly the third) set of critical points are damaged. After the plasticity of the second type (and possibly third type) of failure points following the failure of initial points, the structure is unable to keep its integrity as it was before, as now damage and softening has propagated throughout the whole structure. That is why, in practice, the yielding in the structure usually occurs under external load levels, which are between the external load levels that theoretically cause the first and second set of critical points become locally yielded. This was shown in [36].

### 4.5. Application to Biomedical Implants

There are several applications for cellular materials including heat exchangers, filters, load bearing components, and biomedical implants. Lattice structures can improve the implants' performance significantly, from both mechanical and biological points of view. Obtaining the accurate characteristics and mechanical properties of the lattice structures is necessary to facilitate their use in orthopaedic implants. The mechanical properties of lattice structures depend mainly on the following three parameters: the material they are made of, the cell topology, and the relative density. Obtaining accurate analytical relationships for different cell topologies is a crucial factor in developing computational tools for designing patient-specific implants with non-uniform mechanical property distribution. In this paper, we presented a new and convenient method to transform the analytical relationships of a lattice structure from Euler–Bernoulli theory into Timoshenko theory. Furthermore, using the transformation relationships presented in this paper (Table 1), developing new analytical relationships based on Timoshenko beam theory becomes easier than how it has been before.

For implant design, it is preferable to have a high yield strength and yet access to a wide range of stiffness. Moreover, having similar mechanical properties in three main directions is another factor that should be considered when choosing a unit cell for an implant. This will avoid unwanted deformations generated in the implant when the implant

is placed inside the complex geometrical void it is designed for. The results of this study (Figures 6 and 8) show that among the six unit cells considered, truncated cube followed by diamond and rhombicuboctahedron can best satisfy the above-mentioned characteristics.

*4.6. Limitations of the Present Approach*

Although the conversion methodology and the analytical solutions based on Timoshenko beam theory presented in this study could give very accurate results for elastic mechanical properties of open-cell unit cells and lattice structure, they are still based on linear elastic deformation of structures. Therefore, the analytical relationships are valid only for small deformations (i.e. in the elastic range). The analytical relationships presented in this study could be beneficial for several applications where the deformations remain in the elastic regime, such as in biomedical implants. In some applications of lattice structures and porous materials such as energy absorption and actuation, the matrix material undergoes plastic or hyperelastic deformation, and hence the lattice structures behave non-linearly. Therefore, the range of strains and whether or not the lattice structure's behaviour is linear is one of the most important factors that should be considered before utilization of the analytical relationships presented in this study.

**5. Conclusions**

In this paper, a new methodology to conveniently convert analytical relationships based on Euler–Bernoulli to equivalent Timoshenko ones was presented. Six unit cells for which Euler–Bernoulli analytical relationships could be found in the literature, but Timoshenko theories could not be found were considered: BCC, hexagonal packing, rhombicuboctahedron, diamond, truncated cube, and truncated octahedron [26,30–32,35,39]. The results of this study demonstrated that converting analytical relationships based on Euler–Bernoulli to equivalent Timoshenko ones can decrease the discrepancy between the analytical and numerical values by one order of magnitude, which is a significant improvement in the accuracy of the analytical formulas. The highest improvement in the analytical relationships was observed in the bending-dominated structures such as BCC, diamond, and truncated octahedron topologies and especially at higher relative densities, where the shear deformation effect becomes significant. While the conversion methodology introduced in this study had a significant effect on improvement of elastic modulus and Poisson's ratio relationships, its effect on yield stress relationships was much less, and in some cases (such as BCC, diamond, and truncated cube), it was negligible. The methodology presented in this study is not only beneficial for improving the already-existing analytical relationships, but it also facilitates the derivation of analytical relationships for other, yet unexplored, unit cell types.

**Author Contributions:** Conceptualization, R.H., N.G., M.B., and M.S.; methodology, R.H. and N.G.; software, N.G. and R.H.; validation, N.G., and R.H.; formal analysis, R.H. and N.G.; investigation, R.H., N.G., M.S. and M.B.; resources, R.H. and N.G.; data curation, N.G. and R.H.; writing—original draft preparation, R.H. and N.G.; writing—review and editing, R.H., N.G., M.S., and M.B.; visualization, N.G.; supervision, R.H., M.S. and M.B.; project administration, R.H., N.G., M.S., and M.B. All authors have read and agreed to the published version of the manuscript.

**Funding:** This research received no external funding.

**Institutional Review Board Statement:** Not applicable.

**Informed Consent Statement:** Not applicable.

**Data Availability Statement:** The data presented in this study are available on request from the corresponding author. The data are not publicly available due to their large size and unavailability of storage.

**Conflicts of Interest:** The authors declare no conflict of interest.

**Appendix A. Derivation of Forces and Moments for a Single Strut**

*A.1. Cantilever Beam with Displacement without Rotation in the End*

The governing equations of Timoshenko beam theory are:

$$\begin{cases} EI\dfrac{d^3\varphi}{dx^3} = q(x) \\ \dfrac{dw}{dx} = \varphi - \dfrac{EI}{\kappa AG}\dfrac{d^2\varphi}{dx^2} \end{cases} \tag{A1}$$

Sequential integration from the first line of Equation (A1) gives:

$$EI\frac{d^3\varphi}{dx^3} = q(x) = 0$$

$$EI\frac{d^2\varphi}{dx^2} = C_0$$

$$EI\frac{d\varphi}{dx} = C_0 x + C_1 \tag{A2}$$

$$EI\varphi = \frac{C_0}{2}x^2 + C_1 x + C_2$$

And integration from the second line of Equation (A1) gives:

$$w = \int_0^x \varphi(x)dx - \frac{EI}{\kappa AG}\frac{d\varphi}{dx} + C_3 \tag{A3}$$

There are four boundary conditions at the root and at the end of the beam:

A) @ $x = 0 \rightarrow \varphi = 0$

$$\varphi = \frac{\frac{C_0}{2}x^2 + C_1 x + C_2}{EI} \rightarrow \varphi = \frac{C_2}{EI} = 0 \rightarrow C_2 = 0 \rightarrow \varphi = \frac{C_0 x^2 + 2C_1 x}{2EI} \tag{A4}$$

B) @ $x = L \rightarrow \varphi = 0$

$$\varphi = \frac{C_0 x^2 + 2C_1 x}{2EI} \rightarrow \varphi = \frac{C_0 l^2 + 2C_1 l}{2EI} = 0 \rightarrow C_1 = -\frac{C_0 l}{2} \rightarrow \varphi = \frac{C_0}{2EI}(x^2 - lx) \tag{A5}$$

C) @ $x = 0 \rightarrow w = 0$

$$w = \int_0^x \varphi(x)dx - \frac{EI}{\kappa AG}\frac{d\varphi}{dx} + C_3 \rightarrow w = \frac{C_0}{2EI}\left(\frac{x^3}{3} - l\frac{x^2}{2}\right) - \frac{EI}{\kappa AG}\frac{C_0}{2EI}(2x - l) + C_3$$

$$\rightarrow w = \frac{C_0 l}{2\kappa AG} + C_3 = 0 \rightarrow C_3 = -\frac{C_0 l}{2\kappa AG} \rightarrow w = \frac{C_0}{2EI}\left(\frac{x^3}{3} - l\frac{x^2}{2}\right) - \frac{C_0}{\kappa AG}x \tag{A6}$$

D) @ $x = l \rightarrow w = \delta$

$$w = \frac{C_0}{2EI}\left(\frac{x^3}{3} - l\frac{x^2}{2}\right) - \frac{C_0}{\kappa AG}x \rightarrow w = \frac{C_0}{2EI}\left(\frac{l^3}{3} - l\frac{l^2}{2}\right) - \frac{C_0}{\kappa AG}l = \delta$$

$$\rightarrow C_0 = \frac{-\delta}{\dfrac{l^3}{12EI} + \dfrac{l}{\kappa AG}} \tag{A7}$$

which gives

$$\varphi = \frac{-\delta}{\dfrac{l^3}{6} + \dfrac{2EIl}{\kappa AG}}(x^2 - lx) \tag{A8}$$

which in turn gives

$$w = \frac{-\delta}{\dfrac{l^3}{12EI} + \dfrac{l}{\kappa AG}}\left(\frac{1}{2EI}\left(\frac{x^3}{3} - l\frac{x^2}{2}\right) - \frac{1}{\kappa AG}x\right) \tag{A9}$$

By differentiation from Eq (A9), the point load and at the end of the cantilever beam can be obtained:

A) $@ x = l \rightarrow M = M_0$

$$M(x) = -EI\frac{d\varphi}{dx} \rightarrow M = -\frac{-EI\delta}{\frac{l^3}{6} + \frac{2EIl}{\kappa AG}}(2l - l) = M_0 \rightarrow M_0 = \frac{\delta}{\frac{l^2}{6EI} + \frac{2}{\kappa AG}} \tag{A10}$$

B) $@ x = l \rightarrow Q_x = F$

$$Q_x = \kappa AG\left(-\varphi + \frac{dw}{dx}\right) \rightarrow Q_x = \kappa AG\left(-\frac{EI}{\kappa AG}\frac{d^2\varphi}{dx^2}\right) \rightarrow Q_x = \kappa AG\left(-\frac{EI}{\kappa AG}\frac{-2\delta}{\frac{l^3}{6} + \frac{2EIl}{\kappa AG}}\right)$$

$$= F \tag{A11}$$

$$\rightarrow F = \frac{\delta}{\frac{l^3}{12EI} + \frac{l}{\kappa AG}}$$

*A.2. Cantilever Beam with Rotation without Displacement in the End*

The beam governing equations are the same (Equation (A1–A3)). Applying the relevant boundary condition for this beam:

A) $@ x = 0 \rightarrow \varphi = 0$

$$\varphi = \frac{\frac{C_0}{2}x^2 + C_1 x + C_2}{EI} \rightarrow \varphi = \frac{C_2}{EI} = 0 \rightarrow C_2 = 0 \rightarrow \varphi = \frac{C_0 x^2 + 2C_1 x}{2EI} \tag{A12}$$

B) $@ x = 0 \rightarrow w = 0$

$$w = \int_0^x \varphi(x)dx - \frac{EI}{\kappa AG}\frac{d\varphi}{dx} + C_3 \rightarrow w$$
$$= \frac{1}{2EI}\left(\frac{C_0 x^3}{3} + C_1 x^2\right) - \frac{EI}{\kappa AG}\frac{1}{2EI}(2C_0 x + 2C_1) + C_3 \tag{A13}$$

$$\rightarrow w = -\frac{C_1}{\kappa AG} + C_3 = 0 \rightarrow C_3 = \frac{C_1}{\kappa AG} \rightarrow w = \frac{1}{2EI}\left(\frac{C_0 x^3}{3} + C_1 x^2\right) - \frac{C_0}{\kappa AG}x$$

C) $@ x = l \rightarrow w = 0$

$$w = \frac{1}{2EI}\left(\frac{C_0 x^3}{3} + C_1 x^2\right) - \frac{C_0}{\kappa AG}x \rightarrow w = \frac{1}{2EI}\left(\frac{C_0 l^3}{3} + C_1 l^2\right) - \frac{C_0}{\kappa AG}l = 0$$

$$\frac{C_0 l^3}{6EI} + \frac{C_1 l^2}{2EI} - \frac{C_0}{\kappa AG}l = 0 \rightarrow C_1 = C_0\left(\frac{2EI}{\kappa AGl} - \frac{l}{3}\right) \tag{A14}$$

D) $@ x = l \rightarrow \varphi = \theta$

$$\varphi = \frac{C_0 x^2 + 2C_1 x}{2EI} \rightarrow \varphi = \frac{C_0 l^2 + 2C_1 l}{2EI} = \theta \rightarrow \frac{C_0}{2EI}\left(l^2 + 2l\left(\frac{2EI}{\kappa AGl} - \frac{l}{3}\right)\right) = \theta$$

$$\rightarrow C_0\left(\frac{l^2}{2EI} + \frac{2l}{\kappa AGl} - \frac{l^2}{3EI}\right) = \theta \rightarrow C_0 = \frac{\theta}{\frac{l^2}{6EI} + \frac{2}{\kappa AG}} \tag{A15}$$

E) $@ x = l \rightarrow M = M_0$

$$M(x) = -EI\frac{d\varphi}{dx} \rightarrow M = -EI\left(\frac{1}{2EI}(2C_0 l + 2C_1)\right) = M_0 \rightarrow C_0 l + C_1 = M_0$$

$$\left(\frac{\theta}{\frac{l^2}{6EI} + \frac{2}{\kappa AG}}\right)l + \left(\frac{2EI}{\kappa AGl} - \frac{l}{3}\right)\left(\frac{\theta}{\frac{l^2}{6EI} + \frac{2}{\kappa AG}}\right) = M_0 \rightarrow M_0 = \left(\frac{\frac{2EI}{\kappa AGl} + \frac{2l}{3}}{\frac{l^2}{6EI} + \frac{2}{\kappa AG}}\right)\theta \tag{A16}$$

F) $@ x = l \rightarrow Q_x = F$

$$Q_x = \kappa AG\left(-\varphi + \frac{dw}{dx}\right) \rightarrow Q_x = \kappa AG\left(-\frac{EI}{\kappa AG}\frac{d^2\varphi}{dx^2}\right) \rightarrow Q_x = \kappa AG\left(-\frac{EI}{\kappa AG}\frac{C_0}{EI}\right) = F$$

$$\rightarrow F = -C_0 \rightarrow F = \frac{\theta}{\dfrac{l^2}{6EI} + \dfrac{2}{\kappa AG}}$$

(A17)

## Appendix B. Analytical Equations Extracted from the Literature

**Table A1.** The list of equations extracted from the literature and used in this study

| Eq. Number | Relationship | Equation Number in the Reference | Reference |
|---|---|---|---|
| (A18) | $E = E_x = E_y = E_z = \dfrac{4\sqrt{3}E_o}{\left[\dfrac{l^2}{\pi r^2} + \dfrac{l^4}{2\pi r^4}\right]}$ | (17) | [31] |
| (A19) | $v = v_{xy} = v_{xz} = v_{yx} = v_{yz} = v_{zx} = v_{zy} = \dfrac{-\dfrac{1}{\pi r^2} + \dfrac{l^2}{4\pi r^4}}{\dfrac{1}{\pi r^2} + \dfrac{l^2}{2\pi r^4}}$ | (18) | |
| (A20) | $\delta_{22,b} = \dfrac{FL^3 cos^2\theta}{12E_s I}$ | (7) | [26] |
| (A21) | $\delta_{22,a} = \dfrac{Fsin\theta L}{E_s A} \times sin\theta = \dfrac{FLsin^2\theta}{E_s A}$ | (8) | |
| (A22) | $\delta_{21,b} = \dfrac{Fcos\theta L^3}{12E_s I} \times sin\theta \times sin\dfrac{\pi}{4} = \dfrac{\sqrt{2}FL^3 sin2\theta}{48E_s I}$ | (29) | |
| (A23) | $\delta_{21,a} = \dfrac{Fsin\theta L}{E_s A} \times cos\theta \times sin\dfrac{\pi}{4} = \dfrac{\sqrt{2}FLsin2\theta}{4E_s A}$ | (30) | |
| (A24) | $\dfrac{E}{E_s}$ $= \dfrac{4\pi\left(\dfrac{r}{l}\right)^2}{3(1+\sqrt{2})}\left[\dfrac{4 + 108\left(\dfrac{r}{l}\right)^2 + 207\left(\dfrac{r}{l}\right)^4 + 81\left(\dfrac{r}{l}\right)^6 + \dfrac{G}{E}\left(\dfrac{2}{3} + 19\left(\dfrac{r}{l}\right)^2 + 45\left(\dfrac{r}{l}\right)^4 + 18\left(\dfrac{r}{l}\right)\right)}{8 + 70\left(\dfrac{r}{l}\right)^2 + 105\left(\dfrac{r}{l}\right)^4 + 27\left(\dfrac{r}{l}\right)^6 + \dfrac{G}{E}\left(\dfrac{4}{3} + 13\left(\dfrac{r}{l}\right)^2 + 23\left(\dfrac{r}{l}\right)^4 + 6\left(\dfrac{r}{l}\right)^6\right)}\right]$ | (29) | |
| (A25) | $v = \dfrac{1}{3}\dfrac{8 - 12\left(\dfrac{r}{l}\right)^2 - 36\left(\dfrac{r}{l}\right)^4 + \dfrac{G}{E}\left(\dfrac{4}{3} - \left(\dfrac{r}{l}\right)^2 - 9\left(\dfrac{r}{l}\right)^4\right)}{8 + 70\left(\dfrac{r}{l}\right)^2 + 105\left(\dfrac{r}{l}\right)^4 + 27\left(\dfrac{r}{l}\right)^6 + \dfrac{G}{E}\left(\dfrac{4}{3} + 13\left(\dfrac{r}{l}\right)^2 + 23\left(\dfrac{r}{l}\right)^4 + 6\left(\dfrac{r}{l}\right)^6\right)}$ | (33) | |
| (A26) | $\begin{Bmatrix} 0 \\ 0 \\ 2F \\ 0 \\ 0 \\ 0 \\ 0 \end{Bmatrix}$ | (27) | [30] |

$$= \begin{bmatrix} \frac{96E_sI}{l^3} + \frac{8AE_s}{l}\left(1+\frac{1}{\alpha}\right) & -\frac{8AE_s}{\alpha l} & 0 & 0 & 0 & -\frac{4\sqrt{2}AE_s}{l} + \frac{48\sqrt{2}E_sI}{l^3} \\ \frac{8AE_s}{\alpha l} + \frac{192E_sI}{l^3} + \frac{16AE_s}{l} + \frac{96\sqrt{2}E_sI}{\alpha l^3} & 0 & \frac{8AE_s}{\alpha l} + \frac{192E_sI}{l^3} + \frac{16AE_s}{l} & -\frac{8AE_s}{l}\left(2+\frac{1}{\alpha}\right) - \frac{192E_sI}{l^3} - \frac{96\sqrt{2}E_sI}{\alpha l^3} & -\frac{8AE_s}{\alpha l} & -\frac{48\sqrt{2}E_sI}{l^3} \\ -\frac{8AE_s}{\alpha l} & \frac{8AE_s}{\alpha l} & 0 & 0 & 0 & 0 \\ -\frac{8AE_s}{\alpha l} & 0 & -\frac{8AE_s}{\alpha l} & \frac{8AE_s}{\alpha l} & \frac{8AE_s}{\alpha l} & 0 \\ -\frac{4\sqrt{2}AE_s}{l} - \frac{48\sqrt{2}E_sI}{l^3} - \frac{96E_sI}{\alpha l^3} & 0 & -\frac{96\sqrt{2}E_sI}{l^3} & \frac{96\sqrt{2}E_sI}{l^3} + \frac{96E_sI}{\alpha l^3} & -\frac{4\sqrt{2}AE_s}{\alpha l} & \frac{144E_sI}{l^3} + \frac{4AE_s}{l} + \frac{4AE_s}{\alpha l} \\ & & & & \frac{8AE_s}{\alpha l} & -\frac{4\sqrt{2}AE_s}{\alpha l} \\ \frac{16G_sJ}{\alpha l^2} + \frac{128E_sI}{\alpha l^2} + \frac{32E_sI}{\alpha^2 l^2} + \frac{48\sqrt{2}E_sI}{l^2} & 0 & \frac{48\sqrt{2}E_sI}{l^2} & -\frac{48\sqrt{2}E_sI}{l^2} - \left(\frac{16G_sJ}{\alpha l^2} + \frac{128E_sI}{\alpha l^2} + \frac{32E_sI}{\alpha^2 l^2}\right) & 0 & -\frac{48E_sI}{l^2} \end{bmatrix}$$

(A27)

$$\begin{Bmatrix} 0 \\ 0 \\ 0 \\ F \\ 0 \\ 0 \end{Bmatrix}$$

$$= \begin{bmatrix}
\frac{24E_sI}{l^3}+\frac{2AE_s}{l}+\frac{AE_s}{\alpha l} & -\frac{24E_sI}{l^3}-\frac{2AE_s}{l} & 0 & 0 & -\frac{24EI}{l^3}+\frac{2AE_s}{l} & 0 \\
-\frac{24E_sI}{l^3}-\frac{2AE_s}{l} & \frac{48E_sI}{l^3}+\frac{4AE_s}{l} & -\frac{24E_sI}{l^3}-\frac{2AE_s}{l} & 0 & 0 & 0 \\
0 & -\frac{24E_sI}{l^3}-\frac{2AE_s}{l} & \frac{24E_sI}{l^3}+\frac{2AE_s}{l}+\frac{AE_s}{\alpha l} & -\frac{AE_s}{\alpha l} & \frac{24EI}{l^3}-\frac{2AE}{l} & 0 \\
0 & 0 & -\frac{AE_s}{\alpha l} & \frac{AE_s}{\alpha l} & 0 & 0 \\
-\frac{24E_sI}{l^3}+\frac{2AE_s}{l} & 0 & \frac{24E_sI}{l^3}-\frac{2AE_s}{l} & 0 & \frac{48E_sI}{l^3}+\frac{4AE_s}{\alpha l}+\frac{12AE_s}{l} & -\frac{4AE_s}{\alpha l} \\
0 & 0 & 0 & 0 & -\frac{8AE_s}{\alpha l} & \frac{8AE_s}{\alpha l}
\end{bmatrix} \begin{Bmatrix} \vdots \\ \vdots \\ \vdots \\ \vdots \end{Bmatrix}$$

(36)  [32]

(A28)

$$E_{100} = \frac{\sigma_{zz}}{\varepsilon_z} = \frac{6\sqrt{2}EI}{L^4(1+12I/AL^2)}$$

(15)

[39]

(A29)

$$v_{12} = -\frac{\varepsilon_Y}{\varepsilon_Z} = 0.5\left(\frac{AL^2-12I}{AL^2+12I}\right)$$

(17)

## Appendix C. New Analytical Relationships for Hexagonal Packing Geometry

In this appendix, the derivation of new analytical relationships for the hexagonal packing geometry is presented.

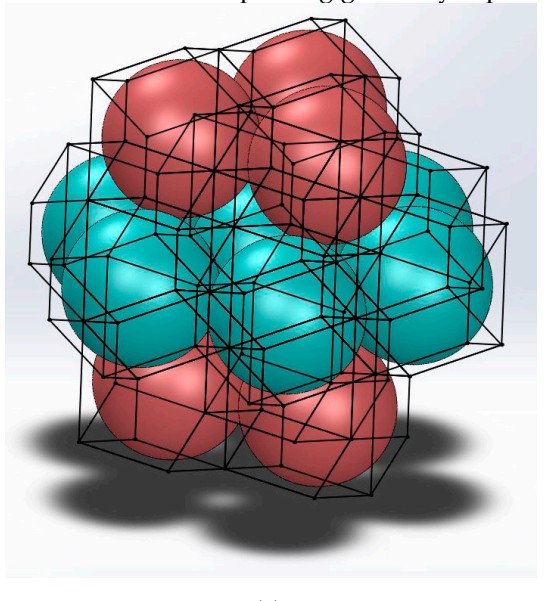

**(a)**

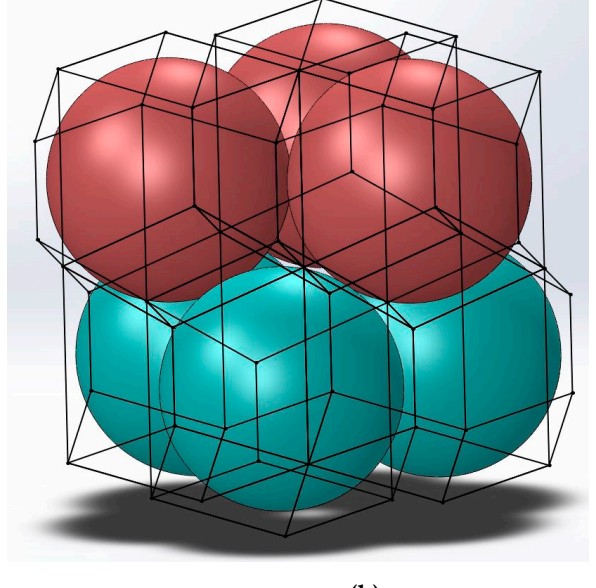

**(b)**

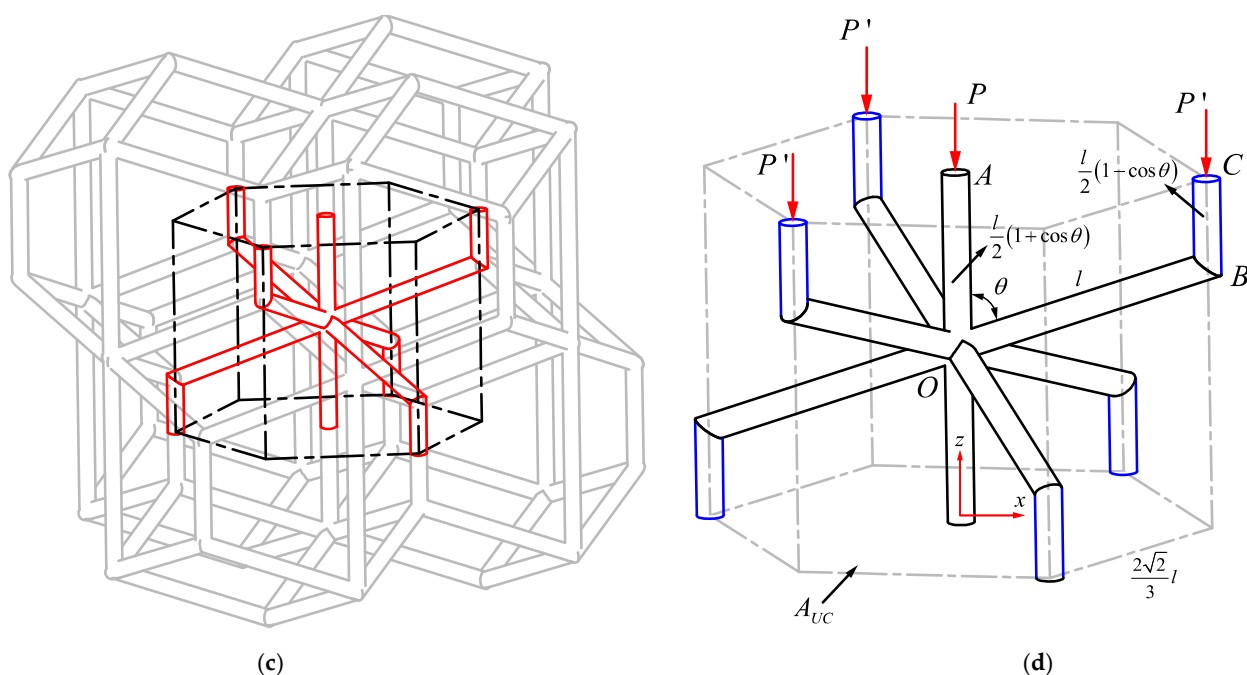

**Figure A1.** (**a**) Spherical hexagonal packing structure; (**b**) Simplified arrangement of spherical hexagonal packing.**;** (**c**) A hexagonal packing unit cell in a lattice structure; (**d**) Dimensions and forces on a hexagonal packing unit cell.

Hexagonal closed-packing (HCP) is known as the mostly efficient way a space can be filled by an arrangement of spheres (Figure A1a). Sphere constructing an HCP structure fill up 74% of space. For an open-cell lattice structure from the concept of HCP, the spheres can be visualized to inflate from all direction until they create a second lattice structure made of polyhedrons (demonstrated by black solid lines in Figure A1a). To find the unit cell of the new polyhedral, first we select two rows of spheres on top of each other and the corresponding polyhedral lattice structure surrounding them (Figure A1b). After removal of the spheres we are left with six polyhedral cells, in which we can find a unit cell which after tessellation in space can create the HCP-based polyhedral lattice structure (the red unit cell highlighted in Figure A1c). Dimensions and loads acting on a hexagonal packing unit cell are shown in Figure A1d. The initial spheres demonstrated in Figure A1a create an angle $\theta$ for which we have $\cos\theta = \frac{1}{3}$ and $\sin\theta = \frac{2\sqrt{2}}{3}$. The cross-sectional area of the hexagonal packing unit cell is therefore $A_{UC} = \frac{3\sqrt{3}}{2} l^2 \sin^2\theta$.

This structure is composed of two types of vertical struts OA (type A) and BC (type C) and one type of inclined strut OB (type B). In each unit cell, there are two type A struts and 6 × 1/3 = 2 type B struts. The term 1/3 is due to the fact that each type B strut is shared by three adjacent unit cells. Displacement of each of the three types of struts are as follows:
i) Type-A strut: Under external load $P$, the change in the length of strut OA is (Equation (2) in [35])

$$\delta_{A,z} = -\frac{Pl}{2AE}(1 + \cos\theta) \tag{A30}$$

ii) Type-C strut: Under external load $P'$, the change in the length of strut BC is (Equation (3) in [35])

$$\delta_{C,z} = -\frac{P'l}{2AE}(1 - \cos\theta) \tag{A31}$$

iii) Type-B strut OB: Displacements of B with respect to O in directions respectively parallel and perpendicular to strut OB are (see Figure 2a,c in the main paper)

$$\delta_B = \frac{P'l}{3AE}\cos\theta \tag{A32}$$

$$\delta'_B = \frac{Ml^2}{6EI} = \frac{P'\sin\theta\, l^3}{36EI}$$

Total displacement of C with respect to O in the z and x directions are respectively:

$$\Delta_{C,z} = \delta_{C,z} + \delta_{B,z} + \delta'_{B,z} = -\frac{P'l}{2AE}(1-\cos\theta) - \frac{P'l}{3AE}\cos^2\theta - \frac{P'l^3}{36EI}\sin^2\theta$$

$$\Delta_{C,x} = \delta_{B,x} + \delta'_{B,x} = -\frac{P'l}{3AE}\sin\theta\cos\theta + \frac{P'l^3}{36EI}\sin\theta\cos\theta \tag{A33}$$

Therefore, the strains in the z and x directions are:

$$\varepsilon_z = \frac{2\delta_{A,z}}{l(1+\cos\theta)} = -\frac{P}{AE}$$

$$\varepsilon_x = \frac{\Delta_{C,x}}{l\sin\theta} = \frac{-\frac{P'l}{3AE}\sin\theta\cos\theta + \frac{P'l^3}{36EI}\sin\theta\cos\theta}{l\sin\theta} = -\frac{P'}{3AE}\cos\theta + \frac{P'l^2}{36EI}\cos\theta \tag{A34}$$

On the other hand, we know that due to continuity of the material and symmetry of each unit cell with respect to the neighbouring cell: $\delta_{A,z} = \Delta_{C,z}$. This means that

$$-\frac{P'l}{2AE}(1-\cos\theta) - \frac{P'l}{3AE}\cos^2\theta - \frac{P'l^3}{36EI}\sin^2\theta = -\frac{Pl}{2AE}(1+\cos\theta)$$

$$\rightarrow \frac{P'}{P} = \frac{1}{\frac{5}{9} + \frac{Al^2}{27I}} \tag{A35}$$

Poisson's ratio can be simply found by

$$v = -\frac{\varepsilon_x}{\varepsilon_z} = \frac{-\frac{P'}{3AE}\cos\theta + \frac{P'l^2}{36EI}\cos\theta}{\frac{P}{AE}} = \frac{-\frac{1}{9} + \frac{Al^2}{108I}}{\frac{5}{9} + \frac{Al^2}{27I}}$$

$$\xrightarrow{circular\ cross-section} v = \frac{-\frac{1}{9} + \frac{1}{27}\left(\frac{l}{r}\right)^2}{\frac{5}{9} + \frac{4}{27}\left(\frac{l}{r}\right)^2} \tag{A36}$$

And normalized elastic modulus can be found by

$$\frac{E_{UC}}{E_s} = \frac{\sigma_z}{E_s\varepsilon_z} = \frac{\frac{P+P'}{A_{UC}}}{\frac{P}{AE_s}E_s} = \left(1+\frac{P'}{P}\right)\frac{A}{A_{UC}} = \left(1+\frac{P'}{P}\right)\frac{A}{\frac{3\sqrt{3}}{2}l^2\sin^2\theta} = \left(1+\frac{1}{\frac{5}{9}+\frac{Al^2}{27I}}\right)\left(\frac{3A}{4\sqrt{3}l^2}\right)$$

$$\xrightarrow{circular\ cross-section} \frac{E_{UC}}{E_s} = \frac{\pi\sqrt{3}}{4}\left(\frac{r}{l}\right)^2\left(1+\frac{1}{\frac{5}{9}+\frac{4}{27}\left(\frac{l}{r}\right)^2}\right) \tag{A37}$$

By defining $S = \frac{AE_s}{l}$ and $T = \frac{12E_sI}{l^3}$, the elastic modulus and Poisson's ratio relationships based on Euler-Bernoulli beam theory for hexagonal packing unit cell are simplified as follows:

$$v = \frac{\frac{S}{T} - 1}{5 + \frac{4S}{T}}$$

$$\frac{E}{E_s} = \frac{\sqrt{3}S\left(1 + \frac{1}{\frac{5}{9} + \frac{4S}{9T}}\right)}{4El} \tag{A38}$$

By substituting $T$ for $\left(\frac{1}{\frac{l^3}{12EI} + \frac{l}{\kappa AG}}\right)$, Poisson's ratio and normalized elastic based on Timoshenko beam theory can be obtained (see Table 1 in the main paper).

## Appendix D. Stress and Strain Contours in the Lattice Structures

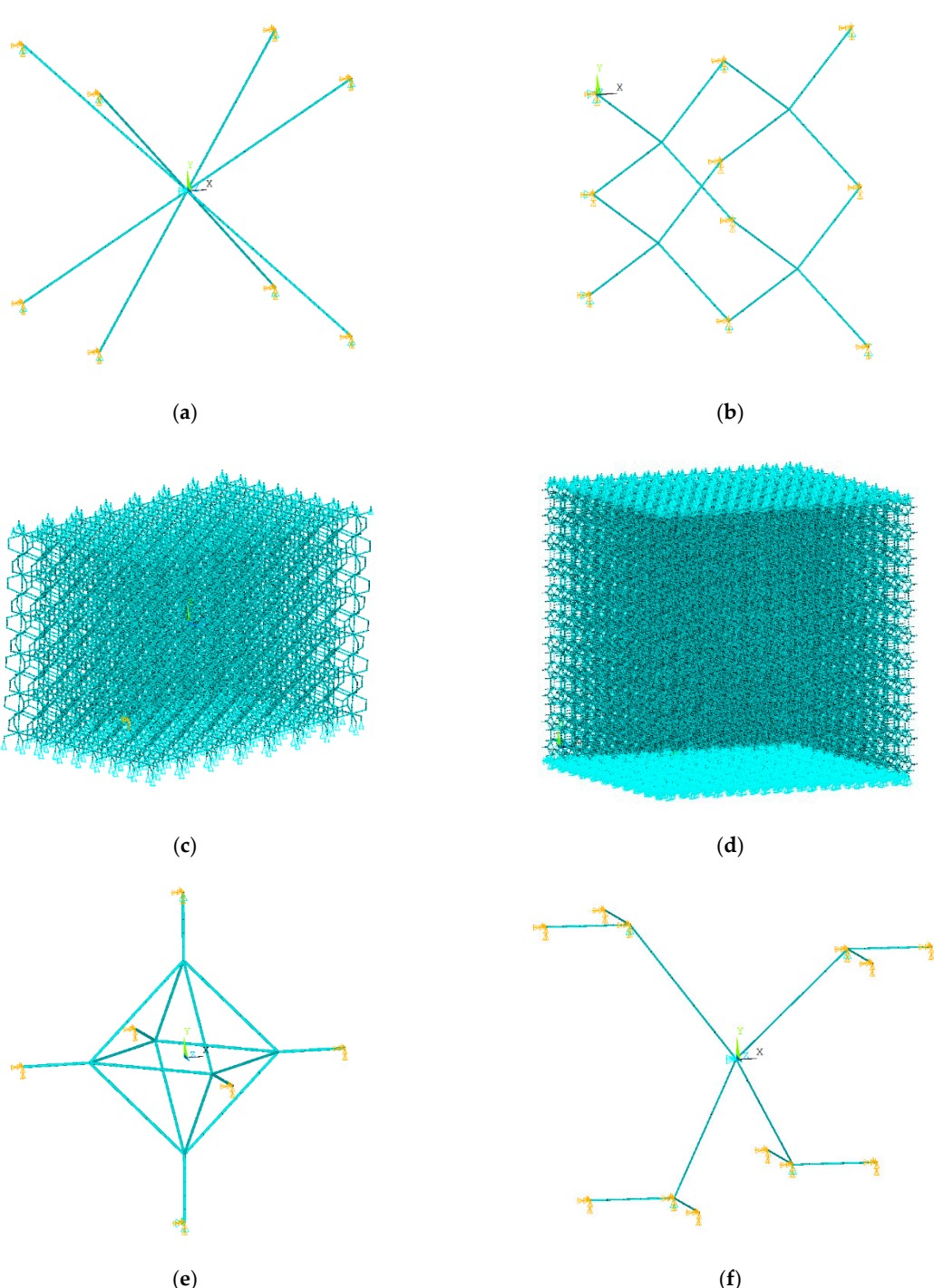

**Figure A2.** FE models and boundary conditions of unit cells: (**a**) BCC; (**b**) diamond; (**c**) hexagonal packing; (**d**) rhombicub-octahedron; (**e**) truncated cube, and (**f**) truncated octahedron in ANSYS APDL.

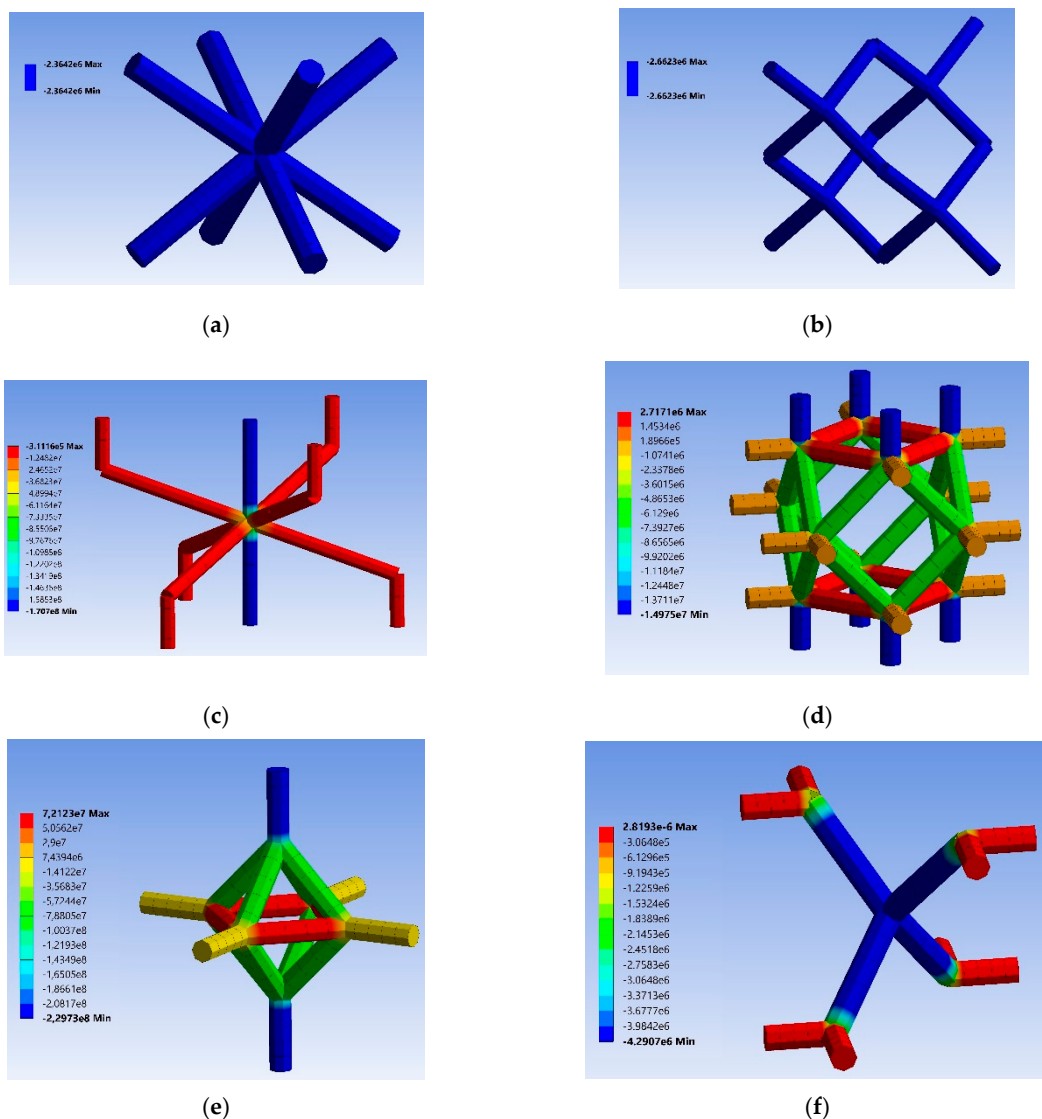

**Figure A3.** Axial stress of unit cells: (**a**) BCC; (**b**) diamond; (**c**) hexagonal packing; (**d**) rhombicuboctahedron; (**e**) truncated cube; and (**f**) truncated octahedron in ANSYS.

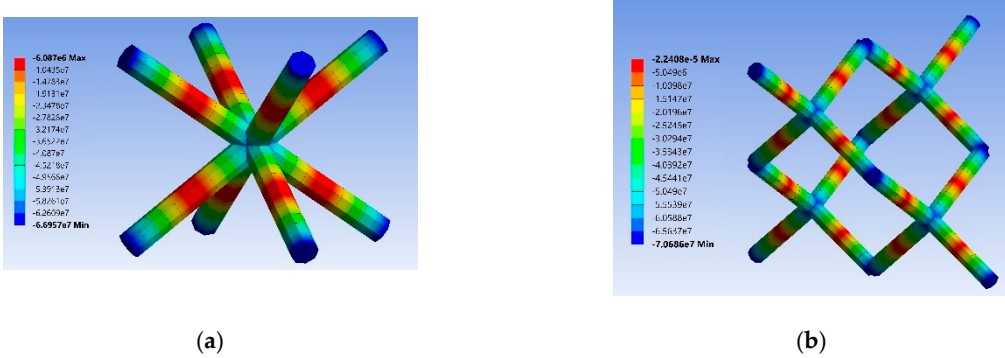

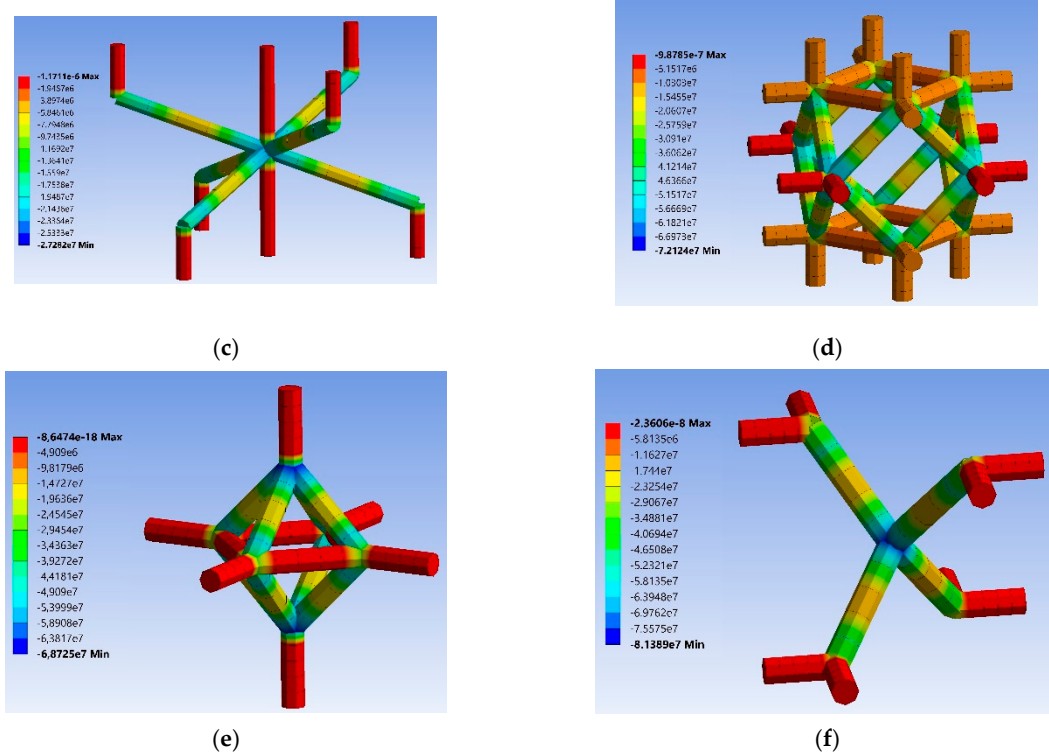

**Figure A4.** Bending stress of unit cells: (**a**) BCC; (**b**) diamond; (**c**) hexagonal packing; (**d**) rhombicuboctahedron; (**e**) truncated cube; and (**f**) truncated octahedron in ANSYS.

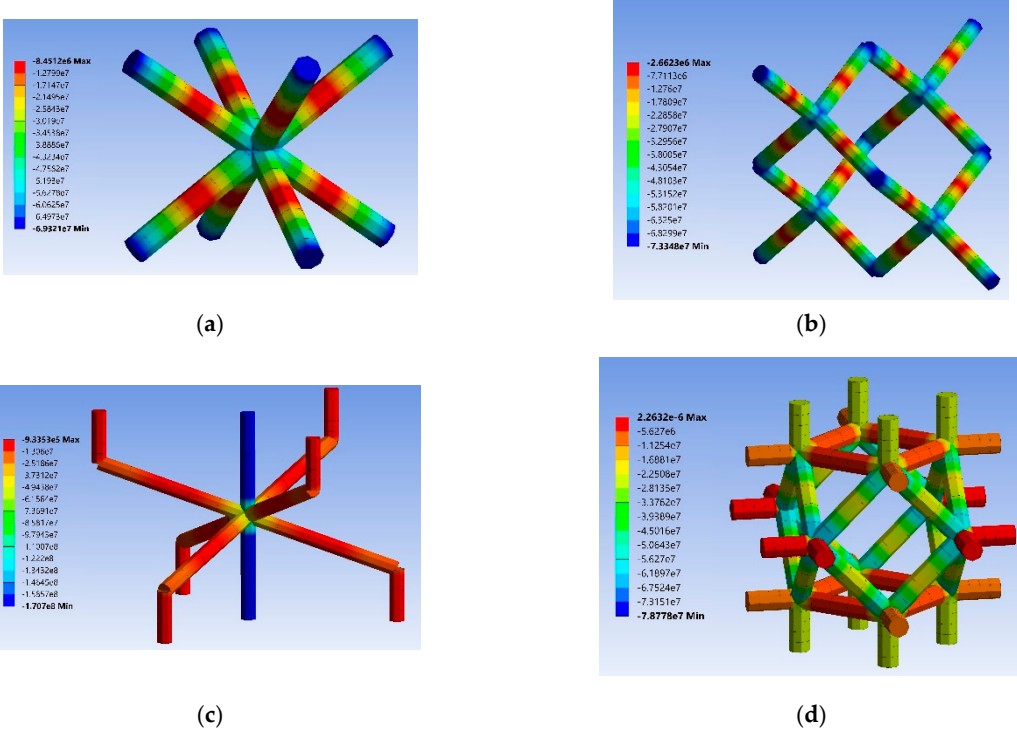

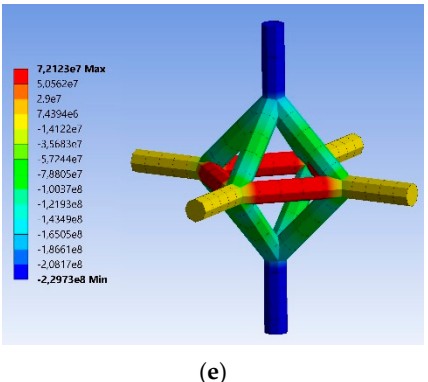
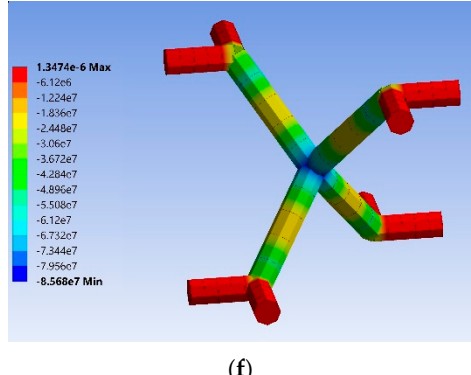

(**e**)                                                                    (**f**)

**Figure A5.** Combined stress of unit cells: (**a**) BCC; (**b**) diamond; (**c**) hexagonal packing; (**d**) rhombicuboctahedron; (**e**) truncated cube, and (**f**) truncated octahedron in ANSYS.

**Appendix E. Effect of Considering Shear Deformation on the Forces/Moments of a Single Strut**

To show the shear deformation effect on the final results of analytical relationships of a single strut, we defined different $\alpha_i$ ratios each being defined as the ratio of the resultant forces and moments (required to create a displacement without rotation or rotation without displacement) in the free end of a cantilever beam based on Euler-Bernoulli to the resultant forces and moments required based on Timoshenko beam theory. The ratios $\alpha_1$, $\alpha_2$ and $\alpha_3$ respectively represent the ratio of the following parameters calculated based on the Euler-Bernoulli and Timoshenko beam theories: the force required for lateral displacement without rotation, $T$, the moment required for lateral displacement without rotation (and the force required to create rotation without displacement), $V$, and the moment required to create rotation without displacement, $U$. The relationships for $\alpha_i$ ratios are derived below:

$$\alpha_1 = \frac{T_{Euler-Bernoulli}}{T_{Timoshenko}} = \frac{\left(\frac{12E_s I}{l^3}\right)}{\left(\frac{1}{\frac{l^3}{12E_s I} + \frac{l}{\kappa A G_s}}\right)} = \frac{\frac{12E_s I}{l^3}}{\frac{1}{\frac{l^3}{12E_s I} + \frac{l}{\kappa A G_s}}} = \frac{12E_s I \left(\frac{l^3}{12E_s I} + \frac{l}{\kappa A G_s}\right)}{l^3} = \frac{l^3 + \frac{12E_s I l}{\kappa A G_s}}{l^3} = 1 + \frac{12E_s I}{\kappa A G_s l^2} \quad \text{(A39)}$$

$$\alpha_2 = \frac{V_{Euler-Bernoulli}}{V_{Timoshenko}} = \frac{\left(\frac{6E_s I}{l^2}\right)}{\left(\frac{1}{\frac{l^2}{6E_s I} + \frac{2}{\kappa A G_s}}\right)} = \frac{\frac{6E_s I}{l^2}}{\frac{1}{\frac{l^2}{6E_s I} + \frac{2}{\kappa A G_s}}} = \frac{6E_s I \left(\frac{l^2}{6E_s I} + \frac{2}{\kappa A G_s}\right)}{l^2} = \frac{l^2 + \frac{12E_s I}{\kappa A G_s}}{l^2} = 1 + \frac{12E_s I}{\kappa A G_s l^2} \quad \text{(A40)}$$

$$\alpha_3 = \frac{U_{Euler-Bernoulli}}{U_{Timoshenko}} = \frac{\left(\frac{4E_s I}{l}\right)}{\left(\frac{\frac{2E_s I}{\kappa A G_s l} + \frac{2l}{3}}{\frac{l^2}{6E_s I} + \frac{2}{\kappa A G_s}}\right)} = \frac{\frac{4E_s I}{l}}{\frac{\frac{2E_s I}{\kappa A G_s l} + \frac{2l}{3}}{\frac{l^2}{6E_s I} + \frac{2}{\kappa A G_s}}} = \frac{4E_s I \left(\frac{l^2}{6E_s I} + \frac{2}{\kappa A G_s}\right)}{l \left(\frac{2E_s I}{\kappa A G_s l} + \frac{2l}{3}\right)} \quad \text{(A41)}$$

The variation of ratios $\alpha_1$, $\alpha_2$ and $\alpha_3$ versus the parameter $r/l$ is presented in the Figure A6. The first important result revealed from the figure is that $\alpha_1$ and $\alpha_2$ are equal for all values of $r/l$. Therefore, considering the shear deformation effect has the same effect on the force required to create lateral displacement without rotation, $T$, on the one hand and moment required to create lateral displacement without rotation (and the force required to create rotation without displacement), $V$, on the other hand. On the other

hand, the shear deformation has smaller effect on $\alpha_3$ in comparison with $\alpha_1$ and $\alpha_2$. Although the $r/l$ is not the ultimate parameter for evaluating the effect of shear deformation in lattice structures for different relative densities, but it is a good measure to predict the difference between the resultant force and moments obtained based on Timoshenko and Euler-Bernoulli beam theories. According to Figure A6, without considering the shear deformation effect in the beam theory, the forces and moments required to create a particular deformation could be predicted by 15–20% higher for $r/l$ as large as 0.15.

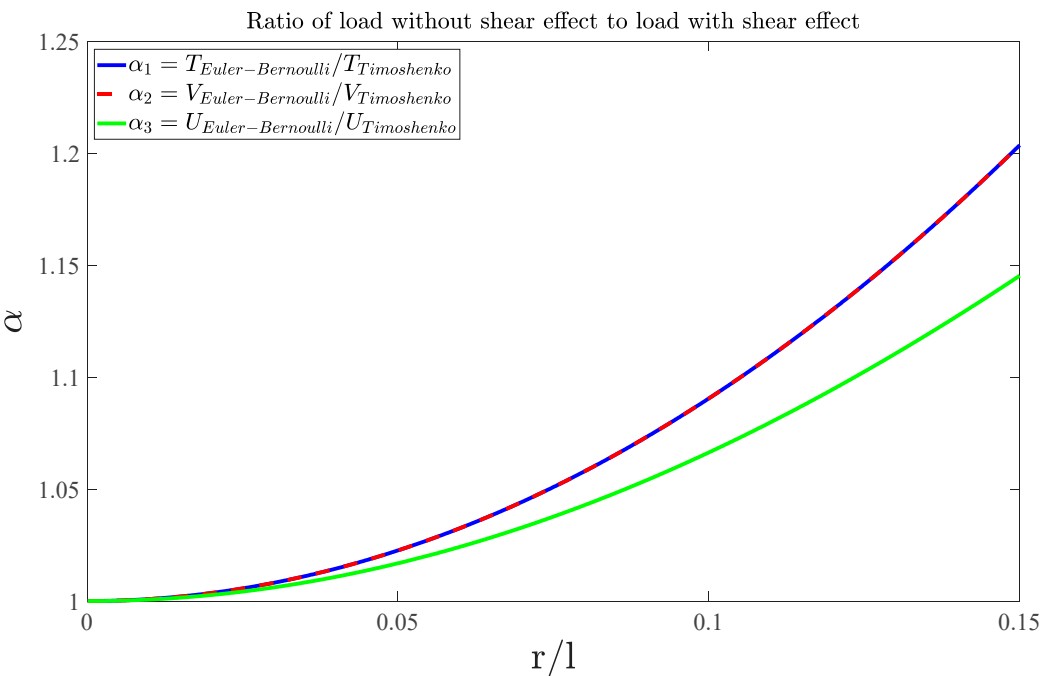

**Figure A6.** The variation of ratios $\alpha_1$, $\alpha_2$ and $\alpha_3$ versus the parameter r/l

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
