# Peer review of "Improving the Accuracy of Analytical Relationships for Mechanical Properties of Permeable Metamaterials"

_applsci, doi:10.3390/app11031332_

Round 1
Reviewer 1 Report
The task of the reviewer would be to formulate recommendations regarding the shape and size of cells of the materials used for the production of implants.
Author Response
Please find the response in attachment

Reviewer 2 Report
The manuscript authored by Hedayati et al. demonstrates a theoretical model for understand the elastic modulus and yield strength of (metallic) open cell foams. Specifically, the authors developed a theoretical model based on Timoshenko beam theory that takes into account the shear effect in the bending deformation of cell walls in an open cell foam. The authors compared the results of their theoretical model with those of numerical simulations. The manuscript can be considered for publication, if the authors address the following issues.
- It is obvious that Timoshenko beam theory will be more appropriate for modeling cell walls in an open cell foam when its relative density becomes large (i.e. the aspect ratio of cell wall becomes large). It would be much better if the authors can show the contributions of both bending deformation and shear deformation to the overall deformation of cell walls with respect to the relative density. In other words, the authors may be able to show the ratio of shear deformation to bending deformation for cell walls during their deformation process.
- The discrepancy between elastic moduli predicted from Euler-Bernoulli beam theory and the authors’ model is critically dependent on the shape of unit cell lattice. The authors should demonstrate how the shape of unit cell lattice determines the deformation process (e.g. shear or tension-dominant deformation vs. bending-dominant deformation) of open cell foam. For example, the inability of Euler-Bernoulli beam theory to predict the effective elastic moduli of open cell foam made based on BCC or diamond unit cells is attributed to the shear-dominant deformation process rather than bending-dominant deformation.
- Can the authors compare their theoretical results (on elastic moduli and yield strengths) for all types of unit cell lattices with the experimental data available in the literature? The authors only showed the comparison of their theoretical results with experimental data only for the case of truncated cube unit cell.
- The authors should cite the book, entitled “Cellular Solids – Structure and Properties” (authored by L.J. Gibson and M.F. Ashby; Cambridge University Press).
Author Response
Please find the response in attachment

Reviewer 3 Report
The article deals with the conversion of existing Euler-Bernoulli analytical relationships to equivalent Timoshenko ones to improve the accuracy of analytical equations for the mechanical properties of repeatable metamaterials, while the technique has been validated at several cell types, so-called lattice structures.
I rate the manuscript positively. It provides a comprehensive view of the issue, and it can be said that the team of authors has an overview of the theories and results achieved by other scientists and experts in this field.
Although I did not find any significant shortcomings, I have a few comments on how to improve it:
- From a methodological point of view, it would be more appropriate for the authors to first present the types of individual cells (as the "Material" in the section "2. Materials and Methods"), and so provide to readers with their characteristics (cell size, strut diameter, ...), ... and only then focus on methods, as the methods refer to the structures.
- From the point of view of better orientation in the text, it would be appropriate for the authors to arrange the pictures so that their order/numbering is in the order mentioned in the text (for example, a reference to Figure 1 is given in the text only after references to Figures 2 and 3, ... it is also similar with references to Figures 2b and 2c).
- The authors should provide a broader discussion regarding the critical points for each unit cell type in Figure 9.
- It would be more appropriate to divide the individual relations given as equation (10) into separate lines.
- References to equations should be enclosed in parentheses in the text, not in quotation marks.
- The conclusions should be formulated more specifically in light of the results achieved.
- On page 7, line 237, it would be appropriate to highlight that these are the Figures a, b, e, f in Annex B, similarly as it is in line 246 on the same page.
As the main text of the article refers to Annexes A to C, this should be a publishable part of it accessible to all readers (it should not be just additional documents provided by reviewers).
Author Response
Please find the response in attachment

Reviewer 4 Report
The authors present analytical relationships based on Euler-Bernoulli to relationships based on Timoshenko beam theory for different lattice structures. These analytical solutions are validated against FE solutions and (some of them) against experimental data from the literature. Overall, the work is interesting, and the methodology may be useful to motivate further works dealing with homogenised constitutive formulations for these materials.
However, it is surprising the quality of the document submitted. It seems that the compilation of the manuscript has been done wrongly. It makes very difficult to review the article. The authors should check the manuscript before final submission. In addition, the writing could be significantly improved by splitting several too long sentences along the manuscript.
Moreover, the authors repeatedly refer to equations from other published works. These expressions, as relevant for the current methods, should be included into an appendix.
Regarding the FE model, are these BC periodic ones? As these models represent an RVE of the lattice, more detailed description about these features must be included.
A discussion on the limitations of the present approach is missing. Several applications of metamaterials are thought to lead to programmed mechanical responses at large deformations. How can these analytical solutions deal with this problem?
Author Response
Please find the response in attachment

Round 2
Reviewer 2 Report
The authors properly addressed all issues that the reviewer raised, and the revised manuscript is acceptable for publication.
Reviewer 4 Report
The authors have addressed my previous concerns.